# Do Vision and Text Cues Exhibit Evidential Coupling?
# UFO: A Benchmark for Compositional Multimodal Reasoning in Unified Models

**Zhongyu Yang** [1]  **Dannong Xu** [1 2 *]  **Yonghan Zhang** [1 *]  **Kefan Chen** [1 *]  **Xinyi Wang** [1 *]
**Yang Xu** [3]  **Wei Pang** [1]  **Yingfang Yuan** [4 1 †]

## Abstract

Unified Foundation Models (UFMs), which support interleaved multimodal generation and understanding, have been proposed as a promising paradigm for reasoning about dynamic world states, yet it remains unclear whether the visual content they generate functions as grounded evidence for subsequent reasoning or merely as auxiliary output. Existing benchmarks largely evaluate generation and understanding as separate capabilities and do not test their functional dependence during reasoning. We introduce **UFO**, a benchmark designed to evaluate whether UFMs generate and use image and text cues as evidence for compositional multimodal reasoning. UFO spans three state-transition regimes, state determination, state reconstruction, and state augmentation, which correspond to progressively smaller transformations of the underlying world state. Our analysis reveals a significant modality gap, as models often achieve high prediction accuracy even when the generated visual cues exert limited influence on their decisions, indicating weakened evidential coupling and a reliance on textual shortcuts rather than robust cross-modal grounding.

## 1. Introduction

**Unified Foundation Models (UFMs)** increasingly integrate multimodal understanding and generation within a single system, driven by the hypothesis that these capabilities are mutually reinforcing. The cornerstone of this integration is the concept of *coupling*. By producing outputs for multiple modalities and tasks in a shared latent space, UFMs are incentivized to maintain cross-modal consistency at inference

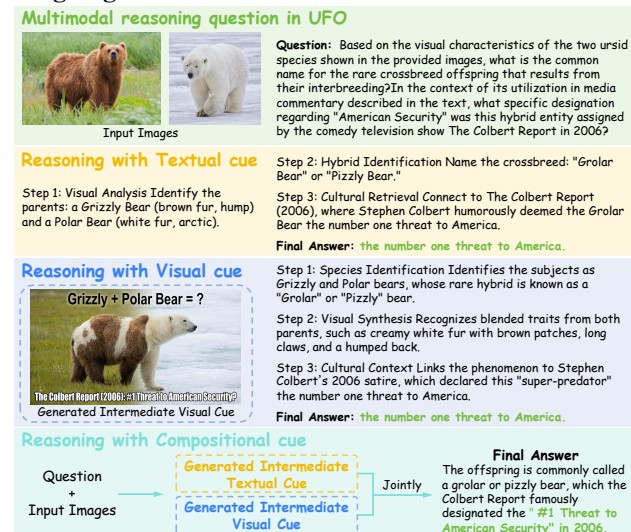

*Figure 1.* **The UFO reasoning framework.** Unlike existing benchmarks that evaluate understanding and generation in isolation, UFO formulates reasoning as a **compositional multimodal process**. Given a question and input images, the model must first generate intermediate *textual and visual cues* representing a future state, and then answer the question by conditioning on these cues. This design enforces **evidential coupling**, requiring intermediate multimodal cues to function as grounded evidence rather than allowing single-step shortcut predictions.

time. From a world-model perspective, coupling is crucial for state-transition reasoning. A model must interpret the current state and construct a plausible future state, supported by complementary evidence across modalities.

However, the impact of coupling on UFM performance remains underexplored, in part because existing benchmarks seldom evaluate coupling directly. In particular, current protocols fall short in three respects. **First**, they largely isolate tasks, treating generation and understanding as separate capabilities rather than as steps in a single inference procedure. **Second**, evaluation is often unimodal. Understanding is probed via text, while generation is assessed via images, without verifying that the two modalities provide aligned, complementary, and grounded evidence. **Third**, benchmarks focus on one-shot predictions, obscuring how evidence is accumulated and used across intermediate steps. Taken together, these gaps suggest that evaluating UFMs as world models requires benchmarks that jointly assess under-

[†]Corresponding author, [*]Equal contribution  [1]BCML, Heriot-Watt University [2]INSAIT [3]Southern University of Science and Technology [4]Northumbria University. Correspondence to: Yingfang Yuan <y.yuan@hw.ac.uk>.

*Table 1.* **Comparison with Existing Multimodal Benchmarks.** We compare UFO with prior benchmarks along seven properties. [1]**Two-step** uses a two-stage procedure in which models first generate intermediate cues and then answer. [2]**Inter. eval.** evaluates intermediate steps or cues rather than only the final prediction. [3]**Multi. metrics** reports multidimensional metrics, beyond a single scalar score, for cue validity. [4]**Hybrid eval.** supports both computational metrics and LLM-assisted evaluation for question answering. [5]**Human labels** includes human annotations for judging or validating intermediate cues. [6]**Contamination** checks for leakage by evaluating without the input image(s). [7]**Filter** screens out examples in which the question can be answered without relying on multimodal cues. UFO is *the only benchmark* that satisfies all criteria and spans 10 task types.

| Benchmark | Venue | Two-step | Inter. eval. | Multi. metrics | Hybrid eval. | Human labels | Contamination | Filtering | MCQ & OQ | #Types |
|---|---|---|---|---|---|---|---|---|---|---|
| ReasonPix2Pix (Jin et al., 2024) | *ArXiv'24* | ✗ | ✗ | ✗ | ✗ | ✗ | ✗ | ✗ | ✗ | 1 |
| ReasonEdit (Huang et al., 2024) | *CVPR'24* | ✗ | ✗ | ✗ | ✗ | ✗ | ✗ | ✗ | ✗ | 1 |
| EditWorld (Yang et al., 2024) | *MM'25* | ✗ | ✗ | ✗ | ✗ | ✗ | ✗ | ✗ | ✗ | 7 |
| Reason50K (He et al., 2025) | *ArXiv'25* | ✗ | ✗ | ✗ | ✗ | ✗ | ✗ | ✗ | ✗ | 4 |
| KRIS-Bench (Wu et al., 2025b) | *NeurIPS'25* | ✗ | ✗ | ✓ | ✓ | ✓ | ✗ | ✗ | ✗ | 7 |
| RISEBench (Zhao et al., 2025) | *NeurIPS'25* | ✗ | ✗ | ✓ | ✓ | ✓ | ✗ | ✗ | ✗ | 4 |
| WorldGenBench (Zhang et al., 2025a) | *ArXiv'25* | ✗ | ✗ | ✓ | ✓ | ✗ | ✗ | ✗ | ✗ | 2 |
| Unified-Bench (Yan et al., 2025) | *ArXiv'25* | ✗ | ✗ | ✗ | ✗ | ✗ | ✗ | ✗ | ✗ | 1 |
| MetaQuery (Pan et al., 2025b) | *ArXiv'25* | ✓ | ✗ | ✗ | ✗ | ✗ | ✗ | ✗ | ✗ | – |
| **UFO (Ours)** | – | ✓ | ✓ | ✓ | ✓ | ✓ | ✓ | ✓ | ✓ | 10 |

standing and generation within explicit state transitions.

To address these limitations, we formulate **compositional multimodal reasoning** for UFMs as a structured inference process over state transitions. Given the current state, a model first generates intermediate multimodal cues that specify the next state, and then answers a question defined over that future state by conditioning on the generated cues.

This formulation enables the study of **evidential coupling**, which is defined as the extent to which the intermediate cues provide grounded, state-consistent evidence that is jointly sufficient for answering questions about the future states. This approach unifies understanding and generation within a single inference trace, spans modalities by producing and consuming visual and textual cues together, and makes the intermediate evidence explicit rather than leaving it hidden inside a single-step prediction.

As summarized in Table 1, prior benchmarks rarely provide the ingredients needed to study evidential coupling. We therefore introduce **UFO** (Unified FOundation), a benchmark for evaluating multimodal reasoning under explicit evidential coupling.

UFO is motivated by a world-state perspective. We organize evaluation into three state-centric settings, formulated with respect to *future* states. **State Determination** infers the future state from current observations. **State Reconstruction** treats the future state as partially specified and requires recovering missing or implicit components. **State Augmentation** treats the future state as insufficiently informative and requires enriching it with state information. Across all settings, we probe evidential coupling by validating the generated multimodal cues and measuring the degree to which they support question answering about future states. Correct answers require consistent evidence (i.e., causal contribution) from both textual and visual cues.

To make coupling interpretable and testable, we adopt a **two-step** protocol that explicitly separates cue generation from answer prediction. We then conduct **intermediate evaluation** to assess whether the generated cues are consistent with the underlying state implied by the inputs. Because coupling can fail in multiple ways, we report **multidimensional metrics** that disentangle *relevance*, *consistency*, *causal utility*, *specificity*, and *compactness*. At scale, **hybrid evaluation** combines lightweight automatic checks with LLM-assisted judging, while **human annotations** are used to calibrate the judges and verify fine-grained cue correctness. To guard against spurious gains, **contamination checks** evaluate models without input images to detect potential information leakage. Finally, **filtering** removes examples that can be solved without relying on multimodal cues.

Our contributions are summarised as follows:

❶ We introduce UFO, a state-centric benchmark for UFMs designed to evaluate two-step reasoning across state determination, reconstruction, and augmentation, with explicit measures of evidential coupling based on cue consistency and causal contribution.

❷ Our experiments indicate that most UFMs fall short of being fully unified, as evidential coupling does not consistently yield performance improvements across state determination, reconstruction, and augmentation tasks in the twelve models evaluated.

❸ We also observe that although the required information gain decreases from state determination through reconstruction to augmentation, this increased ease of evidential coupling does not consistently lead to improved UFM performance.

## 2. Related Work

**Unified Foundation Models.** The rapid progress of MLLMs (Bai et al., 2025; Yang et al., 2026a; 2025b; 2026b;

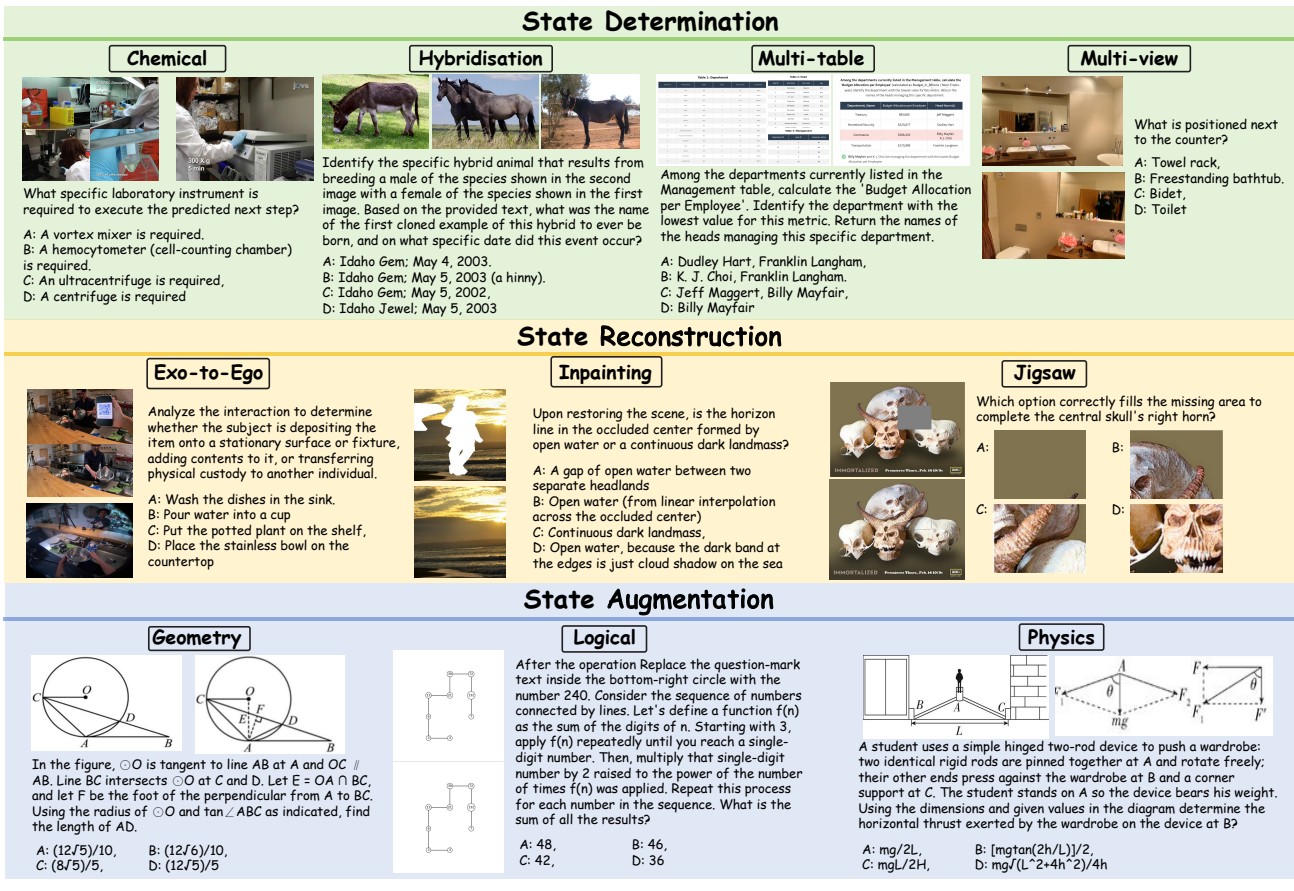

*Figure 2.* The UFO Benchmark for *Compositional Multimodal Reasoning*. UFO evaluates Unified Foundation Models (UFMs) by requiring them to generate *intervenable multimodal cues* as intermediate reasoning steps. The benchmark unifies three state-centric regimes: *State Determination* (inferring latent task-relevant variables), *State Reconstruction* (recovering information from partial observations), and *State Augmentation* (synthesizing auxiliary structural cues). Crucially, UFO moves beyond surface-level alignment to measure *evidential coupling*—quantifying the causal extent to which these intermediate cues are state-consistent and indispensable for the final prediction.

2025a; Liu et al., 2024; Yang et al., 2025c;d; Zhang et al., 2025b; Yang et al., 2025e) has driven the development of *Unified Foundation Models* (UFMs) that integrate multimodal understanding with generation via interleaved text–image interfaces. Early attempts such as Liquid (Wu et al., 2024b), Emu3 (Wang et al., 2024b), and Chameleon (Team, 2024) adopted unified tokenization (Kingma & Welling, 2013; Esser et al., 2021), but exposed an inherent *representation conflict*: tokens optimized for high-fidelity synthesis often fail to preserve the decision-critical semantics required for reasoning. Recent work addresses this tension either by strengthening unified interfaces (Tang et al., 2025; Qu et al., 2025) or, more commonly, by decoupling generation and understanding pathways (e.g., Janus (Ma et al., 2024; Chen et al., 2025b; Wu et al., 2024a), Show-o (Xie et al., 2025), and diffusion-based designs (Pan et al., 2025a; Wu et al., 2025a; Chen et al., 2025a)). These advances raise a central evaluation question: do interleaved intermediate cues encode *task-relevant, intervenable state* that can be monitored and reused across perception, generation, and reasoning, or are they merely plausible artifacts without causal utility?

**Multimodal Reasoning Benchmarks.** Multimodal evaluation has expanded from text-only reasoning (Cobbe et al., 2021; Srivastava et al., 2022; Xu et al., 2026; Hendrycks et al., 2021) to challenging domains including math/science (Lu et al., 2024; Wang et al., 2024a; Li et al., 2024b) and coding (Li et al., 2024a; Chen et al., 2024). However, a persistent validity threat is *modality redundancy*, where models exploit textual shortcuts to reach correct answers without substantive visual processing (Zhang et al., 2024; Yue et al., 2024b; Wang et al., 2025c). While process-oriented benchmarks like ROVER (Liang et al., 2025) and Uni-MMMU (Zou et al., 2025) record intermediate artifacts, they remain fundamentally *observational*: without controlled interventions, they cannot distinguish whether an intermediate cue encodes a **decision-critical state variable** or is a post-hoc hallucination. UFO addresses this limitation by organizing evaluation around state determination, reconstruction, and augmentation, and by introducing cue interventions to test cross-modal state consistency and counterfactual influence.

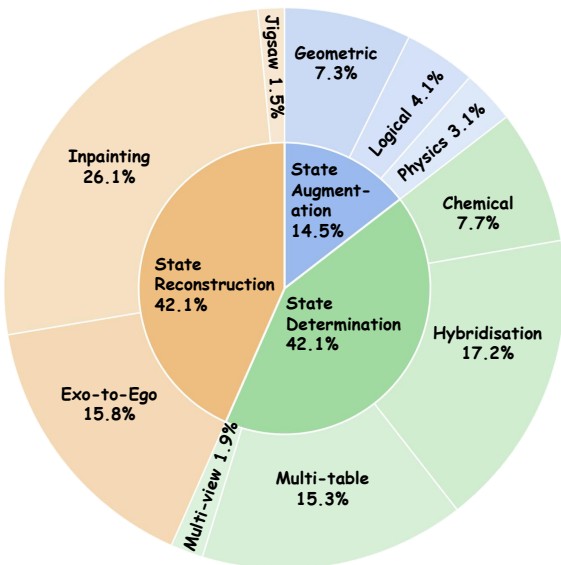

*Figure 3.* **Composition of UFO.**

*Table 2.* **Key statistics of UFO.**

| Statistic | Number | Percentage |
|---|---|---|
| Total questions | 3,936 | |
|   - Multiple-choice questions | 1,943 | 49.4% |
|   - Free-form questions | 1,993 | 50.6% |
|   - Questions with answers | 3,936 | 100.0% |
| Image in the question | 3,936 | 100.0% |
| Problems with single images | 1,182 | 30.0% |
| Problems with multiple images | 2,754 | 70.0% |
| Maximum question length | 756 | |
| Maximum answer length | 70 | |
| Average question length | 44.61 | |
| Average answer length | 2.90 | |

**Gap and Our Positioning.** Taken together, fragmented UFM architectures and observational benchmarks create a blind spot: standard endpoint and similarity metrics cannot diagnose whether intermediate multimodal cues constitute *grounded evidence*—that is, whether they are *state-consistent* and *causally influential*. UFO bridges this blind spot with a **state-centric**, cue-based protocol that keeps the underlying problem structure fixed while intervening on the modality and content of evidence cues. By quantifying cross-modal state consistency and measuring prediction sensitivity under controlled interventions, our framework complements traditional leaderboards with a direct diagnostic of unified, state-consistent reasoning.

## 3. Benchmark

### 3.1. Overview of UFO

**UFO** is a Unified FOundation benchmark for **two-step compositional multimodal reasoning**. Each instance defines a *current* state through the input images and asks a question about a *future* state. A model first generates intermediate *textual and visual cues* that describe this future state. It then answers the question using the generated cues. UFO contains 3,936 questions spanning three types of state transition. These transition types differ in how the future state is constrained by the inputs, enabling us to examine evidential coupling across state determination, reconstruction, and augmentation, corresponding to decreasing levels of information gain. Across all three, the intermediate cues are treated as an explicit description of the future state, and evaluation checks (i) whether the cues are state-consistent and (ii) whether the final answer is justified by those cues.

**State Determination** covers settings in which the future state is fully specified by the conditioning inputs, including the images and the question. All information required to answer the question is explicitly available, and the correct future state follows deterministically from this conditioning. The task therefore reduces to executing the implied state transition, rather than inferring missing content or introducing additional assumptions. This category includes tasks such as *Hybridisation*, where the offspring is uniquely determined by the observed parents; *Chemical*, which requires predicting the next experimental step from the current configuration; *Multi-table*, which involves deriving the relevant target table from multiple provided tables; and *Multi-view*, which requires integrating consistent information across different views of the same scene. These tasks provide a controlled baseline for evidential coupling, as intermediate cues should faithfully encode a uniquely determined future state grounded in the inputs.

**State Reconstruction** arises when the future state is only partially specified by the conditioning inputs. Although the images and the question constrain the state, some required information is absent due to occlusion, corruption, or limited viewpoint. The task is to recover the missing components of the future state while remaining strictly consistent with the available evidence. This category includes *Inpainting*, *Exo-to-Ego*, and *Jigsaw*, where successful reasoning depends on completing unseen regions, reconciling viewpoints, or assembling fragmented observations under strong contextual or geometric constraints. In this setting, evidential coupling tests whether intermediate cues recover information genuinely supported by the inputs, rather than introducing plausible but unsupported completions.

**State Augmentation** addresses cases in which the future state is correctly specified by the conditioning inputs, but answering the question requires making implicit constraints explicit. All entities and relations are observable, yet the raw state description is insufficient for verification, abstraction, or counterfactual reasoning. This category includes

*Table 3.* **Main results.** Comparison of different models' performance on UFO across state determination, reconstruction, and augmentation **multiple-choice question** tasks. The bold font indicates the best performance among the direct, textual, visual, and joint schedules.

| Model | # Params | State Determination | | | | State Reconstruction | | | | State Augmentation | | | | Average | | | |
|---|---|---|---|---|---|---|---|---|---|---|---|---|---|---|---|---|---|
| | | *Direct* | *Textual* | *Visual* | *Joint* | *Direct* | *Textual* | *Visual* | *Joint* | *Direct* | *Textual* | *Visual* | *Joint* | *Direct* | *Textual* | *Visual* | *Joint* |
| *Proprietary Unified MLLMs* | | | | | | | | | | | | | | | | | |
| Gemini-3.0-pro$_{withReasoning}$ | - | 67.55 | 68.23 | 58.84 | 64.76 | 82.44 | 79.41 | 84.13 | 85.31 | 52.32 | 47.60 | 46.23 | 45.42 | 67.65 | 63.90 | 65.31 | 66.13 |
| GPT-5 | - | 45.91 | 46.77 | 48.91 | 51.37 | 52.11 | 51.31 | 49.12 | 53.29 | 41.27 | 46.38 | 48.74 | 50.86 | 46.50 | 48.46 | 49.12 | 52.12 |
| *Open-source Unified MLLMs* | | | | | | | | | | | | | | | | | |
| UniPic1 | 1.5B | 25.91 | 26.40 | **26.64** | 25.79 | **37.12** | 34.02 | 34.54 | 31.83 | **25.97** | **25.97** | 25.26 | 24.91 | **30.74** | 29.61 | 29.84 | 28.26 |
| Ovis-U1 | 2.4 + 1B | 22.56 | **22.80** | 19.72 | 19.82 | **44.11** | 30.00 | 39.00 | 39.22 | 27.72 | 30.88 | 26.67 | **31.23** | **32.59** | 26.99 | 29.03 | 29.84 |
| OmniGen2 | 3B + 4B | 34.03 | 33.33 | **34.30** | 33.92 | 51.52 | 37.41 | 50.36 | **61.18** | **37.19** | 35.44 | 34.74 | 35.44 | 42.02 | 35.39 | 41.28 | **45.88** |
| Janus-Pro | 1B | 26.29 | 25.92 | 26.64 | **27.13** | 32.71 | 35.43 | 35.47 | **37.39** | 23.16 | 23.16 | 25.97 | **27.02** | 28.59 | 29.61 | 30.34 | **31.53** |
| Bagel | 7B MoT | 33.28 | 33.10 | 35.02 | **35.20** | 52.86 | 52.83 | **55.33** | 50.11 | **38.95** | 34.74 | 37.89 | 35.79 | 42.54 | 41.83 | **44.18** | 41.70 |
| EMU3 | 8.5B | 25.18 | 25.06 | 25.06 | **26.16** | **32.32** | 25.84 | 23.24 | 21.16 | **25.26** | 23.15 | 21.40 | 22.11 | **28.26** | 25.12 | 23.74 | 23.41 |
| Omni-R1 | 7B | **25.67** | 24.90 | 24.94 | 25.06 | 28.33 | **29.07** | 28.45 | 29.02 | **24.56** | 20.42 | 22.10 | 23.50 | **26.65** | 26.04 | 26.03 | 26.53 |
| UniCoT | 7B | 34.79 | 35.16 | 35.04 | **37.10** | 43.40 | 40.92 | **43.61** | 39.31 | 37.54 | **39.65** | 38.60 | 37.19 | 38.90 | 38.30 | **39.25** | 38.07 |
| Janus-Pro | 7B | 28.39 | 28.28 | 27.94 | **29.64** | 37.80 | **40.07** | 38.37 | 36.87 | 20.00 | 20.35 | 21.76 | **22.46** | 31.21 | **32.19** | 31.52 | 31.70 |
| UniWorld-V1 | 7B + 12B | 32.48 | 32.97 | 26.64 | **35.28** | 50.91 | **51.08** | 47.72 | 48.48 | 32.98 | **35.44** | 30.18 | 33.68 | 40.49 | **41.12** | 36.23 | 40.73 |
| UniPic2-Metaquery | 9B | 31.42 | 31.01 | 27.48 | **37.74** | **50.70** | 50.62 | 46.06 | 47.65 | 31.23 | 30.88 | 30.18 | **31.93** | 39.69 | 39.43 | 35.87 | **41.15** |

*Geometric*, *Logical*, and *Physics* tasks, which require explicit relations, conditions, or quantities that are not directly expressed in the observations. The task is not to recover missing information, but to introduce auxiliary descriptions that enable principled reasoning.

Taken together, these settings differ in the information gain required to characterise the future state, spanning determination, reconstruction, and augmentation, while uniformly requiring both correct cue generation and answer prediction to assess evidential coupling. The key statistics of UFO are summarised in Table 2 and Figure 3.

### 3.2. Data Curation

Overall, UFO is curated through a two-stage process comprising image collection and question–answer generation, followed by LLM-assisted and human verification. Each instance includes a question paired with one or more images, together with a verified correct answer and corresponding ground-truth cues.

**Stage I: Image Collection and Question–Answer Generation** Based on the world state transition taxonomy, we identify three categories of tasks: state determination, state reconstruction, and state augmentation. We then survey publicly available datasets (Grauman et al., 2024; Yue et al., 2024a; Lin et al., 2014; Bhattad et al., 2025; Xu et al., 2025), wikipedia and select those that naturally support these transition types to construct UFO instances. As data availability varies across tasks, curation procedures differ slightly. We therefore describe the most complex case, Hybridisation, as a representative example, with the remaining tasks following the same general procedure and task-specific adaptations when information is missing or partially specified. In the Hybridisation task, hybridisation is defined as a compositional process in which two parent concepts combine to form a hybrid outcome. This notion extends beyond biological entities to include objects, sports, and cultural artifacts, such

as a spork formed from a fork and a spoon. We use GPT-5.1 to generate candidate hybridisation cases, which are cross-checked using Gemini 3 Pro. For each valid case, we retrieve the corresponding parent and offspring entries from Wikipedia, collecting images of the parents and the hybrid outcome, together with textual descriptions of the offspring. These offspring images and names serve as multimodal cues. Based on this material, we construct two-stage questions using GPT-5.1, first requiring identification of the hybrid outcome and then reasoning about it using the extracted text. To prevent shortcut reasoning, explicit references to parent and offspring names are masked using pronouns.

**Stage II: LM-assisted and Human Verification.** Ensuring question validity is a core design principle of UFO. Accordingly, we employ both human evaluation and LLM-assisted evaluation. Human evaluators verify that each question is properly constructed, with grounded evidential cues and a correct question–answer mapping, while LLM-assisted evaluation is used to detect and mitigate shortcut reasoning arising from legacy dataset biases. To eliminate legacy shortcuts, we regenerate QA pairs from raw media using diverse models (GEMINI-3-PRO, GPT-5.2), retaining only candidates with cross-model consensus. Crucially, we enforce *reasoning necessity* through a dual-layer adversarial stress test. First, to ensure *cue necessity*, we discard instances solvable by external validators (e.g., QWEN2.5-VL, GPT-5.1) via direct QA without intermediate steps. Second, to preclude **data contamination**, we implement a **blind-modality audit**: any instance solvable without visual input is flagged as parametric memorization and strictly excised. This guarantees that UFO evaluates grounded perception rather than latent knowledge or textual bias.

### 3.3. Comparison with Existing Benchmarks

Table 1 highlights a recurring pattern in prior multimodal benchmarks: they improve *individual* aspects of interleaved

Table 4. **Main results.** Comparison of different models' performance on UFO across state determination, reconstruction, and augmentation **open question** tasks. The bold font indicates the best performance among the direct, textual, visual, and joint schedules.

| Model | # Params | State Determination | | | | State Reconstruction | | | | State Augmentation | | | | Average | | | |
|---|---|---|---|---|---|---|---|---|---|---|---|---|---|---|---|---|---|
| | | *Direct* | *Textual* | *Visual* | *Joint* | *Direct* | *Textual* | *Visual* | *Joint* | *Direct* | *Textual* | *Visual* | *Joint* | *Direct* | *Textual* | *Visual* | *Joint* |
| *Proprietary Unified MLLMs* | | | | | | | | | | | | | | | | | |
| Gemini-3.0-pro$_{withReasoning}$ | - | 62.47 | 62.23 | 60.23 | 64.77 | 79.50 | 76.41 | 80.13 | 82.31 | 45.22 | 44.60 | 44.13 | 46.11 | 63.69 | 64.45 | 64.52 | 66.42 |
| GPT-5 | - | 45.35 | 45.95 | 41.41 | 41.25 | 43.86 | 45.96 | 44.12 | 47.21 | 36.38 | 36.38 | 34.11 | 38.13 | 38.13 | 39.14 | 38.12 | 40.33 |
| *Open-source Unified MLLMs* | | | | | | | | | | | | | | | | | |
| UniPic1 | 1.5B | 0.96 | 1.44 | 0.72 | **1.68** | **7.36** | 5.39 | 6.96 | 6.39 | **4.18** | 3.14 | 3.83 | 3.48 | **4.22** | 3.41 | 3.90 | 4.00 |
| Ovis-U1 | 2.4 + 1B | 17.03 | 18.34 | **20.14** | 18.23 | **35.04** | 29.59 | 30.45 | 28.90 | 14.29 | 14.64 | 13.24 | **15.33** | **24.51** | 22.73 | 23.66 | 22.48 |
| OmniGen2 | 3 + 4B | **10.67** | 9.35 | 8.99 | 8.87 | 29.20 | **32.32** | 27.32 | 30.51 | 14.01 | 11.15 | **14.63** | 11.15 | 19.26 | **19.66** | 17.82 | 18.67 |
| Janus-Pro | 1B | 5.40 | 6.04 | **6.42** | 4.34 | 25.78 | **27.43** | 23.02 | 23.65 | 11.50 | 11.50 | **12.54** | 8.01 | 15.20 | **16.19** | 14.56 | 13.32 |
| Bagel | 7B MoT | **8.87** | 8.87 | 8.87 | 7.79 | 36.59 | 31.19 | **37.02** | 29.67 | **16.73** | 14.98 | 15.68 | 13.59 | **22.13** | 19.52 | 22.17 | 18.20 |
| EMU3 | 8.5B | **3.72** | 3.24 | 3.24 | 3.48 | **22.09** | 18.36 | 20.19 | 18.39 | 5.57 | 4.88 | **6.62** | 4.53 | **12.02** | 10.09 | 11.14 | 10.15 |
| Omni-R1 | 7B | 9.32 | 8.35 | 8.41 | **9.80** | 28.72 | **30.92** | 26.31 | 27.64 | 4.83 | **8.29** | 8.05 | 6.54 | 17.16 | **18.22** | 16.19 | 17.13 |
| UniCoT | 7B | **6.95** | 6.84 | 6.48 | 6.71 | 33.22 | 30.26 | **33.53** | 30.42 | 13.94 | 14.98 | **15.33** | 12.89 | 19.45 | 18.26 | **19.59** | 17.98 |
| Janus-Pro | 7B | 5.16 | **5.40** | 4.79 | 5.16 | **29.24** | 26.71 | 26.77 | 25.45 | 9.06 | **11.85** | 10.80 | 10.45 | **16.26** | 15.65 | 15.27 | 14.80 |
| UniWorld-V1 | 7B + 12B | 26.14 | 26.14 | 25.64 | **27.28** | **34.55** | 30.13 | 31.13 | 29.76 | 14.29 | 13.94 | **14.64** | 12.20 | **28.11** | 26.13 | 26.46 | 26.19 |
| UniPic2-Metaquery | 9B | 10.02 | **10.26** | 9.30 | 9.18 | **32.82** | 31.97 | 29.44 | 31.80 | 11.50 | 11.85 | **12.54** | 11.50 | **20.21** | 19.99 | 18.58 | 19.41 |

reasoning (e.g., exposing interleaving or enforcing visual dependence), yet evaluation remains largely *observational*. As a result, intermediate cues can be fluent and well-aligned while remaining non-essential to the final decision. UFO reframes this by shifting the objective from "*getting the answer right*" to "*using the right state for the right reason*". This causal perspective drives three key differentiators:

**From Visual Dependence to Process Verification.** Benchmarks that emphasize visual dependence ensure images matter in principle, but they rarely test *how* evidence is used along the way. UFO makes intermediate cues first-class objects via Process Evaluation: by enforcing cue necessity during curation and evaluating intermediate steps rather than just endpoints, success implies genuine state derivation rather than shortcut heuristics or post-hoc narration. **From Subject Collections to Regime-Level Diagnosis.** Standard subject-based groupings (e.g., Math, Physics) aggregate heterogeneous skills, thereby obscuring *why* models fail. UFO instead organizes tasks by **state operation**: determination, reconstruction, and augmentation. This regime-centric design turns a leaderboard into a diagnostic instrument, effectively localizing failures to distinct mechanisms (e.g., misextracted state vs. failed recovery) that topic-level scores cannot disentangle on their own.

**From Holistic Accuracy to Multi-Source Cue Disentanglement.** Standard benchmarks typically evaluate reasoning as a monolithic process, often masking the provenance of the solution. UFO disrupts this by structurally decomposing performance into four distinct inference pathways: *text-cued*, *image-cued*, and *joint-cued* reasoning, benchmarked against a *direct answer* baseline. This granular factorization allows us to isolate the specific contribution of each modality. Instead of merely checking the final output, we further verify whether success stems from genuine cross-modal synergy (where Joint > Unimodal) or reliance on single-modality priors, ultimately offering a precise audit of the model's reasoning dependencies.

## 4. Experiment

### 4.1. Experimental Setup

**Models.** We evaluate twelve representative UFMs in a zero-shot setting, including two proprietary models (Gemini-3.0-pro (Deepmind, 2026) and GPT-5.2 (OpenAI, 2026)) and ten open-source models. Based on their design paradigms and generation mechanisms, we group the open-source UFMs into three architectural categories:

- **Pure autoregressive unified MLLMs.** These models employ a single autoregressive transformer trained with a unified next-token objective to support both multimodal understanding and image generation under one interface. A representative example is Emu3 (Wang et al., 2024b).[1]

- **Hybrid unified models with a dedicated image generator.** These models combine an autoregressive reasoning backbone with a dedicated image generator, often diffusion- or DiT-based, to enhance visual fidelity while preserving strong instruction following and controllability. This group includes Ovis-U1 (Wang et al., 2025a), UniPic2-Metaquery (Wei et al., 2025), and OmniGen2 (Wu et al., 2025a).

- **Autoregressive unified models with architectural specialization.** These models remain autoregressive but introduce explicit specialization between understanding and generation, such as decoupled visual encodings, expert routing, or task-specific masking, while maintaining a unified interaction interface. This group includes Janus-Pro (Chen et al., 2025b), UniPic1 (Wang et al., 2025b), Bagel (Deng et al., 2025), Omni-R1 (Cheng et al., 2026), and UniCoT.

---

[1]Some autoregressive unified models further introduce architectural specialization (e.g., decoupled visual encodings or expert routing). For clarity, we group such variants separately below.

**Metrics:** We use classification accuracy as the primary evaluation metric for all main experiments. Accuracy is appropriate in our setting since all benchmarks are formulated as single-answer tasks with unambiguous ground-truth.

## 4.2. Implementation Details

All experiments are conducted on NVIDIA H100 GPUs (80GB) using bfloat16 precision. To ensure fair comparison across models and pruning strategies, we fix the inference batch size and decoding hyperparameters for all settings, including temperature (0.2), top-$k$ (1), diffusion timesteps (100), and text-guided scale (7.5). All remaining hyperparameters and implementation details strictly follow the official configurations released by the respective models.

## 4.3. Main Results

**State-Centric Stratification and the Generative Gap.** Consistent with the state-centric regimes defined in our framework, we observe a distinct performance hierarchy across Tables 3 and 4: *State Reconstruction > State Augmentation > State Determination*. This stratification empirically validates the theoretical difficulty implied by the state transitions: *State Reconstruction*, which involves recovering information from partial observations, benefits from tighter constraints on the hypothesis space, allowing models like OmniGen2 to reach 61.18% accuracy (MCQ). Conversely, *State Determination* proves most challenging, as it requires inferring latent future variables from current observations with high entropy. Crucially, evaluating these transitions reveals a severe *discriminative-generative gap*. While models maintain serviceable baselines in multiple-choice settings, performance collapses in open-ended generation (e.g., UniPic1 drops from 25.91% to 0.96% in Determination). This suggests that despite the promise of unified modeling, current UFMs often rely on surface-level discrimination rather than constructing the robust, grounded world models required for explicit state generation.

**The Challenge of Evidential Coupling.** The central hypothesis of UFMs, that multimodal capabilities are mutually reinforcing, is challenged by our findings on *evidential coupling*. While intermediate cues (textual or visual) individually outperform direct inference in specific regimes, their combination does not consistently yield the expected additive gains. In grounded tasks like *State Reconstruction*, visual cues effectively serve as complementary evidence (e.g., OmniGen2 improves from 51.52% Direct to 61.18% Joint). However, in the more abstract *State Determination* and *Augmentation* regimes, the Joint schedule frequently trails the strongest single-modality baseline (e.g., UniCoT: 39.65% Textual vs. 37.19% Joint). These results indicate that simply producing outputs in a shared latent space does not guarantee effective coupling; instead, heterogeneous

| Model | # Params | State Determination | | State Reconstruction | | State Augmentation | |
|---|---|---|---|---|---|---|---|
| | | *Textual* | *Visual* | *Textual* | *Visual* | *Textual* | *Visual* |
| *Proprietary Unified MLLMs* | | | | | | | |
| Gemini-3.0-pro | - | 41.42 | 59.17 | 33.52 | 71.51 | 22.55 | 38.49 |
| GPT-5 | - | 44.96 | 47.13 | 24.91 | 72.41 | 51.90 | 12.45 |
| *Open-source Unified MLLMs* | | | | | | | |
| UniPic1 | 1.5B | 17.31 | 4.72 | 11.41 | 3.22 | 7.70 | 5.31 |
| Ovis-U1 | 2.4 + 1B | 12.35 | 6.31 | 9.50 | 4.43 | 12.10 | 2.30 |
| OmniGen2 | 3 + 4B | 19.13 | 4.21 | 17.5 | 5.22 | 11.1 | 5.19 |
| Janus-Pro | 1B | 22.13 | 2.34 | 9.31 | 4.10 | 10.11 | 0.88 |
| Bagel | 7B MoT | 21.30 | 17.41 | 19.31 | 13.55 | 22.47 | 11.41 |
| EMU3 | 8.5B | 12.35 | 5.71 | 10.39 | 7.21 | 14.12 | 4.66 |
| Omni-R1 | 7B | 12.80 | 8.08 | 14.52 | 6.90 | 14.13 | 3.84 |
| UniCoT | 7B | 20.93 | 8.99 | 15.32 | 11.11 | 15.50 | 7.41 |
| Janus-Pro | 7B | 17.99 | 13.45 | 14.46 | 9.04 | 10.63 | 3.11 |
| UniWorld-V1 | 7B + 12B | 13.13 | 9.74 | 9.73 | 7.41 | 16.22 | 4.12 |
| UniPic2-Metaquery | 9B | 17.91 | 13.99 | 13.76 | 4.12 | 13.30 | 7.63 |

*Table 5.* **Process-level cue accuracy.**

cues can introduce interference or compete for attention. Consequently, achieving true evidential coupling, where intermediate cues function as jointly sufficient, grounded evidence, remains a non-trivial obstacle for current unified architectures.

## 4.4. Process Evaluation

To explain the limited gains from evidential coupling, Table 5 evaluates the quality of intermediate cues, revealing a severe imbalance between textual and visual fidelity. Open-source models consistently articulate state transitions better than they instantiate them visually; for instance, the 1B Janus-Pro model achieves 22.13% textual accuracy versus only 2.34% for visual synthesis in State Determination. This disparity provides a structural explanation for the modality interference observed earlier, as low-quality visual cues act as high-entropy noise that disrupts rather than supports the inference process. Furthermore, the generally low absolute performance highlights a critical bottleneck in formulating UFMs as world models. While models can partially articulate future states in text, their ability to instantiate these predictions into a consistent visual reality remains nascent. Even for proprietary models like Gemini-3.0-pro, textual accuracy peaks at around 41%, indicating that the internal simulation of future states is far from perfect. These findings confirm that achieving effective evidential coupling requires more than shared latent spaces. It demands a fundamental improvement in the fidelity of intermediate generation to ensure that self-generated cues serve as reliable, high-quality supports for downstream compositional reasoning.

## 5. VLM as Judge

To establish a scalable and reliable evaluation pipeline, we construct a human-annotated validation set using intermediate cues generated by GEMINI-3-PRO and GPT-5.2. We employ this dataset to rigorously benchmark candidate VLM judges and scoring protocols, ensuring our automated metrics align with human consensus.

| Judge Model | Visual-cue | | Text-cue | | Avg. | |
|---|---|---|---|---|---|---|
| | Acc. ↑ | κ ↑ | Acc. ↑ | κ ↑ | Acc. ↑ | κ ↑ |
| *Proprietary MLLMs* | | | | | | |
| GPT-4o-mini | 63.3 | 16.5 | 41.9 | 22.4 | 52.6 | 19.5 |
| GPT-4o | 66.5 | 23.6 | 33.8 | 31.4 | 50.2 | 27.5 |
| GPT-5.1 | 61.6 | 23.4 | 37.9 | 28.3 | 49.8 | 25.9 |
| Gemini-2.5-Flash | 63.0 | 16.7 | 37.7 | 23.3 | 50.4 | 20.0 |
| Gemini-2.5-Pro | 64.4 | 22.1 | 38.0 | 22.8 | 51.2 | 22.5 |
| Gemini-3-Flash | 68.6 | 25.1 | 57.3 | 22.7 | 63.0 | 23.9 |
| *Open-source MLLMs* | | | | | | |
| Gemma-3-12B | 56.6 | 10.9 | 48.8 | 20.0 | 52.7 | 15.5 |
| Gemma-3-27B | 65.8 | 18.6 | 62.2 | 18.2 | 64.0 | 18.4 |
| Qwen2.5-VL-32B | 50.5 | 13.8 | 69.7 | 18.7 | 60.1 | 16.3 |
| Qwen2.5-VL-72B | 55.1 | 15.6 | 44.1 | 31.0 | 49.6 | 23.3 |
| Qwen3-VL-8B | 63.1 | 20.2 | 45.5 | 17.8 | 54.3 | 19.0 |
| Qwen3-VL-32B | 69.1 | 27.2 | 46.4 | 26.6 | 57.8 | 26.9 |
| Qwen3-VL-30B-A3B | 59.6 | 14.9 | 53.6 | 21.8 | 56.6 | 18.4 |
| Qwen3-VL-235B-A22B | 60.6 | 19.7 | 47.2 | 28.1 | 53.9 | 23.9 |

*Table 6.* Alignment between candidate VLM judges and human labels under visual-cue and text-cue judging settings, categorized by proprietary and open-source models.

| Judge Model | Methods | Visual-cue | | Text-cue | | Avg. | |
|---|---|---|---|---|---|---|---|
| | | Acc. ↑ | κ ↑ | Acc. ↑ | κ ↑ | Acc. ↑ | κ ↑ |
| Qwen3-VL-32B | Binary | 69.1 | 27.2 | 46.4 | 26.6 | 57.8 | 26.9 |
| | Tournament | 59.2 | 19.2 | 39.2 | 23.0 | 49.2 | 21.1 |
| | Majority Voting | 59.4 | 18.6 | 46.5 | 31.5 | 53.0 | 25.1 |
| | Chain-of-Thought | 57.5 | 20.0 | 49.1 | 25.1 | 53.3 | 22.6 |
| | Confidence Weighted Voting | 59.0 | 18.5 | 47.3 | 24.2 | 53.2 | 21.4 |
| Gemini-3-Flash | Binary | 68.6 | 25.1 | 57.3 | 22.7 | 63.0 | 23.9 |
| | Tournament | 50.5 | 15.2 | 49.9 | 24.1 | 50.2 | 19.7 |
| | Majority Voting | 47.6 | 11.7 | 59.0 | 29.1 | 53.3 | 19.7 |
| | Chain-of-Thought | 72.8 | 28.9 | 62.0 | 31.2 | 67.4 | 30.1 |
| | Confidence Weighted Voting | 49.5 | 13.6 | 58.3 | 24.4 | 53.9 | 19.0 |

*Table 7.* Alignment between candidate VLM judge methods and human labels under visual-cue and text-cue judging settings.

## 5.1. Meta-Evaluation of Judges

We identify the most capable judge models by benchmarking alignment with human annotations in Table 6. Among proprietary models, GEMINI-3-FLASH demonstrates the highest consistency, achieving a Kappa ($\kappa$) of 25.1 in visual evaluation. Notably, in the open-source landscape, QWEN3-VL-32B emerges as superior, outperforming significantly larger counterparts with a peak visual $\kappa$ of 27.2. This indicates that parameter count does not strictly correlate with judging capability, which is likely due to better instruction-following in specific checkpoints.

Building on these selections, Table 7 further investigates optimal judging methodologies. We observe a modality-dependent preference: visual cues are robustly evaluated using Binary scoring, whereas textual cues, involving subtle semantic nuances, benefit significantly from Chain-of-Thought (CoT) reasoning. For instance, applying CoT to Qwen3-VL-32B improves textual alignment stability compared to binary scoring. Thus, we adopt QWEN3-VL-32B using a hybrid protocol (Binary for visual, CoT for text) as our primary open-source evaluation strategy.

| Model | Params | Relevance | Faithfulness | Causal Util. | Specificity | Compactness | Average |
|---|---|---|---|---|---|---|---|
| *Proprietary Unified MLLMs* | | | | | | | |
| Gemini-3.0-pro | - | 3.47 | 3.43 | 2.94 | 3.10 | 4.12 | 3.41 |
| GPT-5 | - | 3.55 | 3.31 | 2.40 | 2.88 | 3.77 | 3.19 |
| *Open-source Unified MLLMs* | | | | | | | |
| UniPic1 | 1.5B | 1.48 | 2.35 | 1.45 | 1.43 | 1.75 | 1.69 |
| Ovis-U1 | 2.4 + 1B | 0.98 | 1.34 | 0.55 | 1.30 | 1.12 | 1.06 |
| OmniGen2 | 3 + 4B | 1.50 | 2.42 | 1.45 | 2.13 | 1.31 | 1.76 |
| Janus-Pro | 1B | 1.32 | 1.12 | 0.93 | 1.36 | 1.04 | 1.15 |
| Bagel | 7B MoT | 2.41 | 2.58 | 2.19 | 2.47 | 2.19 | 2.38 |
| EMU3 | 8.5B | 1.79 | 1.92 | 1.35 | 2.65 | 1.79 | 1.90 |
| Omni-R1 | 7B | 1.54 | 1.43 | 1.71 | 2.18 | 2.13 | 1.80 |
| UniCoT | 7B | 1.32 | 2.41 | 1.39 | 1.77 | 2.11 | 1.80 |
| Janus-Pro | 7B | 1.92 | 2.03 | 1.31 | 1.64 | 1.70 | 1.72 |
| UniWorld-V1 | 7B + 12B | 2.03 | 2.00 | 1.34 | 2.04 | 1.11 | 1.70 |
| UniPic2-Metaquery | 9B | 2.01 | 2.16 | 1.95 | 2.53 | 2.31 | 2.19 |

*Table 8.* **Evidence-oriented evaluation of visual cue generated.**

## 5.2. Multi-dimensional Cue Analysis

To diagnose the breakdown of evidential coupling, we evaluate cue quality across five dimensions: *Relevance*, *Faithfulness*, *Causal Utility*, *Specificity*, and *Compactness* (rubrics in Appendix A.2.2). The results in Table 8 reveal a critical *semantic gap*: models consistently score higher on surface-level *Relevance* than on reasoning-centric *Causal Utility*. This suggests current UFMs function primarily as *associative retrievers* rather than grounded *world simulators*: producing generic scenes that are thematically related but lack the specific "decision hinge" required to resolve state uncertainty. Furthermore, universally low *Specificity* scores indicate that generated evidence is typically ambiguous or illustrative rather than decisive. Consequently, these vague visual outputs act as semantic noise. Instead of reinforcing the inference trace through grounded coupling, they dilute the model's attention, providing an explanation for the modality interference observed in our main results.

## 6. Conclusion

This work examines whether Unified Foundation Models effectively couple generation and understanding when reasoning over world states. We introduce UFO, a state centric benchmark for two-step compositional multimodal reasoning across state determination, reconstruction, and augmentation. Each instance includes question answering pairs with ground truth multimodal cues, allowing direct evaluation of whether intermediate cues are state consistent and causally relevant. Our results show that evidential coupling varies across models and tasks and does not consistently improve performance. Using human labeled cues and a vision language model as a judge, we further find that textual and visual contributions are often misaligned and can even degrade accuracy. By separating cue generation from answering, UFO exposes limitations of evaluation protocols that collapse reasoning into a single prediction and provides a diagnostic framework emphasising state consistency and causal contribution beyond aggregate accuracy metrics. We believe that UFO provides the community with a principled and comprehensive framework for exploring and evaluating Unified Foundation Models.

## Impact Statement

Unified Foundation Models are increasingly expected to reason about hypothetical, future, or counterfactual states in domains where intermediate decisions matter. However, prevailing evaluation protocols often reduce reasoning to a single prediction, making it difficult to distinguish genuine evidence-based inference from shortcut behavior. By separating cue generation from answering and explicitly validating the causal role of intermediate multimodal evidence, UFO offers a principled means of diagnosing this distinction.

The primary impact of UFO lies in its ability to reframe how multimodal reasoning systems are evaluated rather than in optimizing any particular model architecture. By emphasizing state consistency, intervenability, and causal contribution of intermediate cues, the benchmark encourages the development of models whose reasoning processes are more transparent and auditable. We expect UFO to serve as a diagnostic tool for the community, enabling more fine-grained analysis of multimodal reasoning behaviors that are otherwise obscured by aggregate accuracy metrics.

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

## Overview of the Appendix

# A. Implementation Details

### A.1. Model Architectures

**Emu3** (Wang et al., 2024b) Emu3 employs a pure autoregressive architecture trained with a unified next-token prediction objective. By interleaving discrete visual codes (via a visual quantizer) with text tokens, it performs both understanding and generation within a single shared parameter space, eliminating the need for separate encoders or diffusion models.

**Ovis-U1** (Wang et al., 2025a) Ovis-U1 utilizes a hybrid architecture that structurally aligns visual embeddings with language representations. It integrates an LLM with a dedicated image generator through a probabilistic mapping mechanism, leveraging the reasoning backbone for semantic guidance while decoding visual outputs via a specialized generator.

**OmniGen2** (Wu et al., 2025a) OmniGen2 combines an autoregressive reasoning backbone with a diffusion-based generation module. It focuses on optimizing the interface between instruction following and visual synthesis by decoding the LLM's latent representations into structural control signals, thereby balancing semantic controllability with generative fidelity.

**UniPic2-Metaquery** (Wei et al., 2025) UniPic2-Metaquery introduces a "Metaquery" mechanism within a hybrid framework to handle complex multimodal instructions. It explicitly separates reasoning from pixel-level synthesis, allowing the LLM to formulate structured queries that plan content layout before delegating rendering to a dedicated generator.

**Janus-Pro** (Chen et al., 2025b) Janus-Pro resolves the conflict between perception and generation through decoupled visual pathways. While sharing a single autoregressive transformer, it uses a SigLIP encoder for high-level semantic understanding and a separate VQ-tokenizer for fine-grained generation, ensuring optimal performance for both tasks without interference.

**UniPic1** (Wang et al., 2025b) UniPic1 serves as a foundational autoregressive unified model, processing concatenated sequences of textual and visual tokens via a single transformer. It employs a standard VQ-VAE for visual tokenization, prioritizing a seamless unified interaction interface where understanding and generation are treated as equivalent sequence modeling tasks.

**Bagel** (Deng et al., 2025) Bagel addresses task interference through architectural specialization, such as Mixture-of-Transformer-Experts (MoT) or task-specific routing. By dynamically allocating parameters for different modalities, it manages the gradient conflicts between understanding and generation losses, preserving LLM reasoning depth while enabling robust visual synthesis.

**Omni-R1** (Cheng et al., 2026) Omni-R1 enhances the reasoning capabilities of unified models by integrating reinforcement learning strategies. It focuses on a "plan-then-generate" approach, ensuring that generated visual content is logically consistent with the multi-step deductions derived by the autoregressive backbone.

**UniCoT** (Qin et al., 2025) UniCoT integrates Chain-of-Thought (CoT) reasoning into the generative process. It trains the model to produce intermediate textual rationales before synthesizing visual tokens, thereby grounding image generation in explicit logical deductions and improving performance on multi-hop reasoning tasks.

**UniWorld-V1** (Lin et al., 2025) UniWorld-V1 emphasizes world modeling by treating video or image sequences as state transitions. It employs a unified architecture trained on large-scale predictive tasks, learning to simulate future physical states based on current observations to serve as a predictive engine for dynamic environments.

### A.2. Prompt

#### A.2.1. PROMPTS FOR RESPONSE GENERATION

> **Textual Cue Generation Prompt**
>
> My specific question is: {Question}
> Your task is to generate a key textual cue based on the provided reference information. This textual cue should:
> Identify key entities, attributes, and relationships pertinent to the query. Focus on critical visual features (e.g., structure, spatial layout, states) that drive reasoning. Ensure the response is concise and informative, limited to 1-2 sentences.
> The generated text cue should serve as an abstract, language-only substitute for a visual cue, helping to answer the question above.

---

**Visual Cue Generation Prompt**

My specific question is: {Question}
Your task is to generate an image based on the provided reference images. This generated image should serve as a visual cue to help answer the question above.

---

A.2.2. PROMPTS FOR EVALUATION GENERATION

**Multi-dim Cue Evaluation Prompt**

**Role.** You are a strict evaluator of a model-generated Visual CoT (visual cue image) for a Visual QA task. Your goal is to judge whether the generated cue provides *usable visual evidence* to support the ground-truth answer, beyond answer correctness.
**Inputs (ONLY these):**

- Question: $Q$

- Ground-truth Answer: $A_{\text{gt}}$

- Generated Visual CoT image: $I_{\text{gen}}$

**What to evaluate.** Score the evidential quality of $I_{\text{gen}}$ under five dimensions: (1) Evidence Relevance, (2) Faithfulness / Logical Consistency, (3) Causal Utility, (4) Localization / Specificity, (5) Non-redundancy / Compactness. Each dimension is scored as an **integer from 0 to 5**.
**Hard rules.**

- Judge evidence, not aesthetics.

- Only reward information that is *visually verifiable* in $I_{\text{gen}}$.

- If a factor is implied but not visually checkable, treat it as missing or weak.

- Penalize hallucinated or distracting content that could alter the decision.

**Step 1: Identify decision-critical factors.** From $Q$ and $A_{\text{gt}}$, extract the minimal set of visual factors required to justify the answer (e.g., object identity, attribute, count, spatial relation, readable text). Mark each factor as **critical** (answer would flip if wrong) or **supporting**.
**Step 2: Verify factors in $I_{\text{gen}}$.** For each factor, assign one status: `verified` (clearly visible), `weak` (ambiguous/partial), `missing` (not shown), or `contradicted` (opposite evidence shown).
**Step 3: Identify hallucinations or distractors.** List any visually salient content that is not required by $Q$ but may mislead the answer.
**Output format (STRICT JSON ONLY).**

```
{
  "decision_critical_factors": [
    {"factor":"...", "criticality":"critical|supporting",
     "status":"verified|weak|missing|contradicted", "why":"..."}
  ],
  "scores": {
    "relevance": 0,
    "faithfulness": 0,
    "causal_utility": 0,
    "specificity": 0,
    "compactness": 0
  },
  "failure_modes": [
    "missing_critical_evidence",
    "ambiguous_evidence",
    "contradiction_to_answer",
    "misleading_hallucination",
    "generic_scene"
  ],
  "notes": "Brief justification based only on visible evidence."
}
```

## Multi-dim Cue Evaluation Scoring Rubrics Prompt

1. **Evidence Relevance.**
   *What:* Coverage of **decision-critical visual factors** required to justify $A_{gt}$ (only visually verifiable evidence counts).
   *Scoring (0–5):*

   - **0**: No decision-critical evidence.
   - **1**: Thematic similarity only; no factor verifiable.
   - **2**: Few factors weakly supported; most missing.
   - **3**: Several factors verified, but at least one **critical** factor missing or ambiguous.
   - **4**: Most critical factors clearly verified; minor gaps only.
   - **5**: All or nearly all critical factors clearly and directly verified.

   *Rule:* If any critical factor is missing or contradicted, relevance $\leq 3$.

2. **Faithfulness / Logical Consistency.**
   *What:* Logical compatibility with $Q$ and $A_{gt}$; absence of contradictions or **misleading hallucinations**.
   *Scoring (0–5):*

   - **0**: Contradicts $A_{gt}$ or strongly supports an alternative answer.
   - **1**: Major inconsistency or hallucination likely flipping the decision.
   - **2**: Inconsistency affecting a critical factor.
   - **3**: Minor hallucinations, not affecting the decision.
   - **4**: Fully consistent; only trivial extra details.
   - **5**: Fully consistent; no competing or misleading content.

3. **Causal Utility (counterfactual proxy).**
   *What:* Expected reduction of uncertainty relative to using $Q$ alone; presence of a clear **decision hinge**.
   *Scoring (0–5):*

   - **0**: Misleading or harmful.
   - **1**: No added evidence beyond the question.
   - **2**: Weak confirmation; answer largely guessable.
   - **3**: Meaningful support; uncertainty reduced.
   - **4**: Clear decisive evidence resolving the ambiguity.
   - **5**: Uniquely decisive evidence; cue is essential for confident resolution.

   *Rule:* If no critical factor is clearly verifiable, causal utility $\leq 2$.

4. **Localization / Specificity.**
   *What:* Precision and inspectability of evidence (not a generic illustration).
   *Scoring (0–5):*

   - **0**: Generic scene; queried evidence not checkable.
   - **1**: Evidence implied only.
   - **2**: Evidence present but poorly visible or ambiguous.
   - **3**: Decisive evidence visible, partial ambiguity remains.
   - **4**: Clear and specific; low ambiguity.
   - **5**: Unmistakable, sharply localized decisive evidence.

5. **Non-redundancy / Compactness.**
   *What:* Signal-to-noise ratio; focus on necessary evidence without competing distractors.
   *Scoring (0–5):*

   - **0**: Dominated by distractors or misleading content.
   - **1**: Heavy clutter; evidence hard to isolate.
   - **2**: Noticeable distractors competing with evidence.
   - **3**: Some redundancy; evidence still salient.
   - **4**: Clean composition; minimal distraction.
   - **5**: Maximally compact; only decision-critical evidence shown.

### A.3. Licenses

Table 9 summarizes the licensing information of all models evaluated in this work. We rely exclusively on publicly released research models with official project pages or model cards that specify their usage terms. Several systems, including Ovis-U1, Omni-Gen2, Janus-Pro, Bagel, EMU3, and UniCoT, are distributed under the Apache-2.0 license, enabling broad academic use of their implementations. Other models, such as UniPic1, UniPic2-Metaquery, and UniWorld-V1, provide dedicated license files or usage statements within their repositories, which we follow as specified by the original authors. For Omni-R1, we adhere to the research-only usage

*Table 9.* License information for models evaluated in this paper.

| Model | Official URL | License / Terms |
|---|---|---|
| UniPic1 | Project Page | MIT License |
| Ovis-U1 | Project Page | Apache-2.0 License |
| OmniGen2 | Project Page | Apache-2.0 License |
| Janus-Pro | Project Page | MIT License |
| Bagel | Project Page | Apache-2.0 License |
| EMU3 | Project Page | Apache-2.0 License |
| Omni-R1 | Model Card | MIT License |
| UniCoT | Project Page | Apache-2.0 License |
| UniWorld-V1 | Project Page | MIT License |
| UniPic2-Metaquery | Project Page | MIT License |

terms described in its official model card. Across all cases, models are used solely for non-commercial academic evaluation, and no proprietary, closed-access, or restricted commercial resources are included.

## B. Example of Data

Figures B.1–B.10 collectively reveal a systematic gap in how current multimodal benchmarks assess reasoning. Across all ten cases, the dominant failure mode is not a lack of perceptual recognition or linguistic competence, but the inability to construct *intermediate evidence* that is both state-consistent and causally actionable.

In process-driven domains such as chemistry and physics (Figs. B.1, B.10), correct answers are only reachable when latent transition variables or causal trajectories are explicitly encoded in the cue. Models that bypass this step can still appear competent under one-shot evaluation, yet fail immediately once the reasoning is decomposed. Similarly, in symbolic and relational settings, including geometry, logical graphs, and multi-table reasoning (Figs. B.3, B.7, B.8), the decisive factor is whether the cue exposes hidden structure, enabling subsequent operations to be performed on a well-defined intermediate state rather than on memorized patterns.

Vision-centric tasks further stress the necessity of evidential coupling. Hybridisation, multi-view, and exo-to-ego reasoning (Figs. B.4, B.9, B.2) demonstrate that visually plausible generations are insufficient if the produced cues fail to preserve discriminative attributes aligned with the textual constraint. Likewise, reconstruction-based tasks such as inpainting and jigsaw reasoning (Figs. B.5, B.6) expose models that recover visually coherent content but omit the specific spatial or compositional evidence required for answering.

Taken together, these cases show that conventional answer-only accuracy systematically overestimates reasoning ability by conflating *shortcut inference* with evidence-based reasoning. UFO directly targets this blind spot by enforcing a two-step protocol and validating the causal contribution of intermediate multimodal cues. As a result, the benchmark does not merely ask whether a model can answer correctly, but whether it can *justify that answer through grounded, intervenable evidence*. This property is essential for treating UFMs as reliable world models rather than black-box predictors.

## Chemical reasoning question in UFO

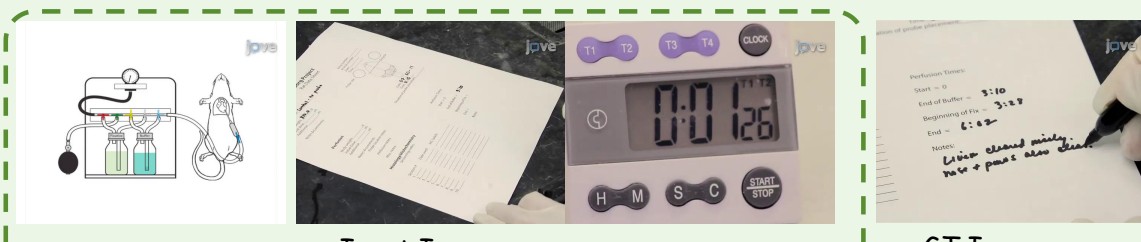

Input Images                    GT Image cue

**Question:** If the predicted next step shown in the protocol is omitted, what specific reaction property can no longer be calculated?

A: The reaction order with respect to the limiting reagent,
B: The average perfusion/flow rate through the system (mL/min),
C: The activation energy (Ea) via an Arrhenius analysis",
D: The reaction rate or kinetics.

Answer: D

Ground Truth Text Cue: Record completion time

## Reasoning with Textual cue

**text_cue_generated_gpt:**
If the protocol omits the predicted "next step" (the subsequent elementary reaction/transition state), you cannot calculate the reaction rate/kinetics—specifically the activation energy (ΔG‡) and the associated rate constant—because the energy barrier between current and next states is undefined.

**text_cue_generated_gemini:**
Based on the provided images, the key textual cue is:The visual evidence features a digital timer and a data sheet with specific fields for recording the start and end times of the perfusion steps, highlighting the precise monitoring of the procedure's duration. This temporal measurement is essential for calculating the flow rate of the fluids administered during the reaction.,

## Reasoning with Visual cue

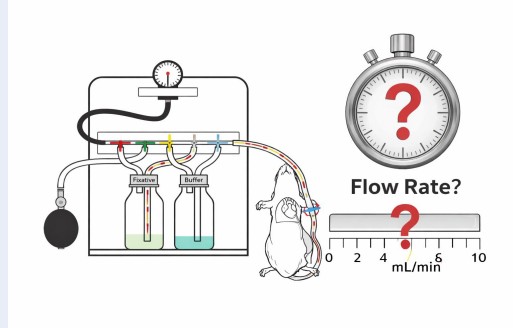

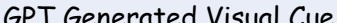

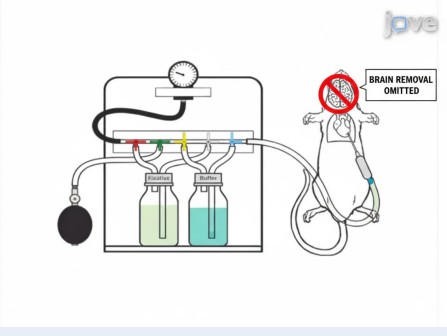

GPT Generated Visual Cue                    Gemini Generated Visual Cue

*Figure B.1.* **Case 1: Chemical reasoning (state determination).** The question is answerable only when the intermediate cue correctly encodes the task-relevant reaction attribute, illustrating how cue quality governs downstream answerability under the two-step protocol.

## Exo-to-Ego reasoning question in UFO

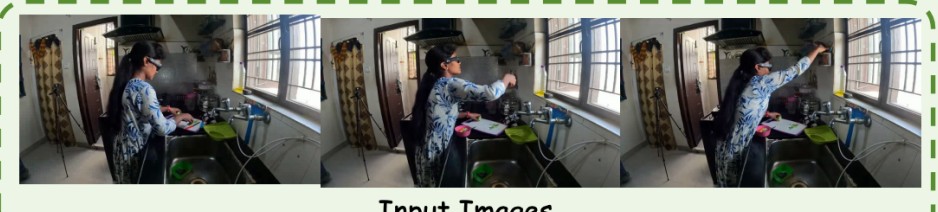

Input Images

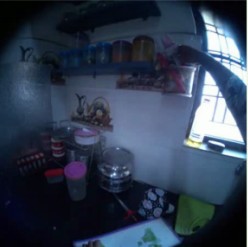

GT Image cue

**Question:** Analyze the physical points of contact and object interactions in the scene to infer what task is being performed and how the action unfolds.

A: Placing cups on the rack,
C: Put away the peeler in the utensil basket,

B: Cutting vegetables on a board,
D: Pouring water into a cup

Answer: C

Ground Truth Text Cue: The scene shows a vegetable-chopping task that requires two-handed coordination: the left arm (identified by a pink watch strap) stabilizes the vegetable on the cutting board while the right hand grips the knife or reaches for items. Both hands are actively engaged, making the task inherently bimanual rather than one-handed.

## Reasoning with Textual cue

**text_cue_generated_gpt:**
Determine which hand(s) directly touch and manipulate the target object: note contact points (fingers/palm) and whether one hand alone grips/presses while the other is off-object. If both hands simultaneously hold/steady opposite sides or perform coordinated actions, classify as both hands.

**text_cue_generated_gemini:**
The left hand stabilizes the vegetable on the cutting board, while the right hand grips the knife to slice it.

## Reasoning with Visual cue

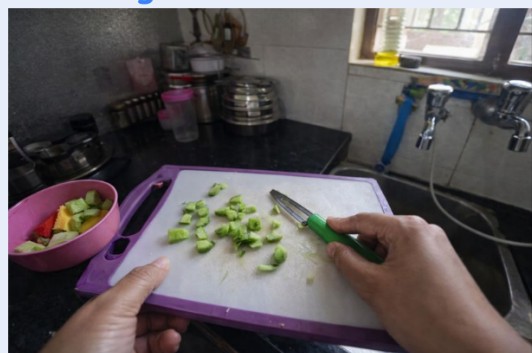
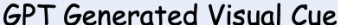

GPT Generated Visual Cue

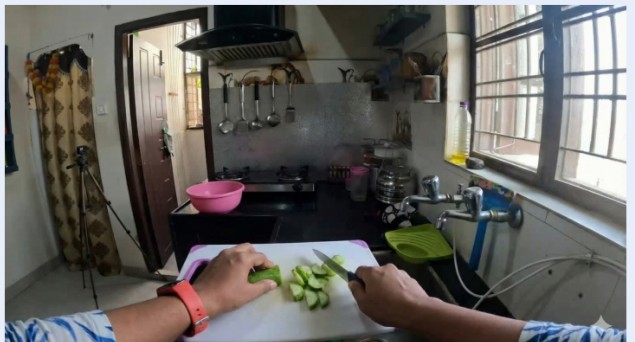

Gemini Generated Visual Cue

*Figure B.2.* **Case 2: Exo-to-ego reasoning (viewpoint-conditioned action).** The instance requires grounding the decision in physical contact and hand-object interaction, where subtle visual evidence must be carried into the cue to support a correct classification.

## Geometry reasoning question in UFO

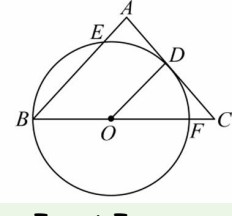

Input Images

**Question:** In △ABC with ∠A = 90°, points O and D lie on BC and AC respectively and satisfy \\(CD \\cdot BC = AC \\cdot OC\\). With center O and radius OD, draw a circle passing through B, intersecting AB at E and BC at F. Using the given segment lengths marked for AE and CF, find the inradius \\(r\\) of △ABC.

A: 10        B:8        C:12        D:9

Answer: D

Ground Truth Text Cue: Draw OH⊥AB through point O, and draw the vertical foot as H

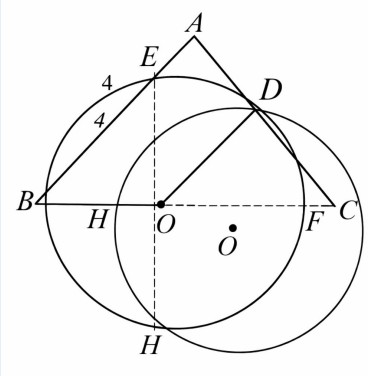

GT Image cue

## Reasoning with Textual cue

**text_cue_generated_gpt:**
Right △ABC with ∠A=90° (AB ⊥ AC); O on hypotenuse BC, D on AC with condition CD·BC = AC·OC. Circle centered at O with radius OD passes through B, meets AB at E and BC at F; drop OH ⊥ AB. Given AE=4 and CF=5, find inradius r.

**text_cue_generated_gemini:**
The image depicts a right-angled triangle $ABC$ with a circle centered on the hypotenuse $BC$ that passes through vertex $B$ and is tangent to leg $AC$ at point $D$. Key geometric features include the construction of $OH \\perp AB$, which forms a rectangle with the radius $OD$ and bisects the chord on $AB$, thereby connecting the given segment lengths $AE=4$ and $CF=5$ to the circle's radius.

## Reasoning with Visual cue

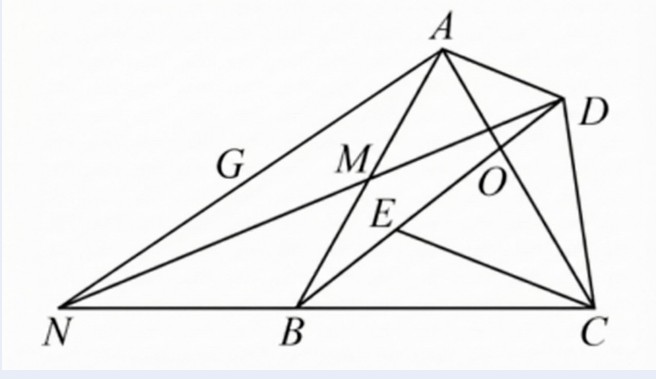

GPT Generated Visual Cue          Gemini Generated Visual Cue

*Figure B.3.* **Case 3: Geometry reasoning (constructive invariants).** The cue must specify the missing auxiliary construction (e.g., a perpendicular drop) that makes the solution trace executable, turning implicit geometric structure into explicit intermediate evidence.

## Hybridisation reasoning question in UFO

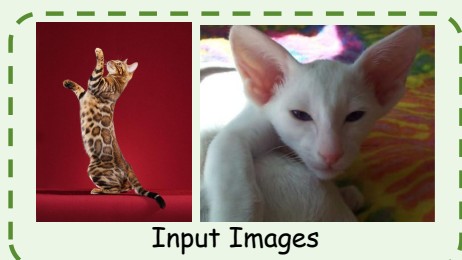
Input Images

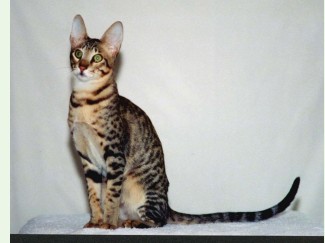
GT Image cue

**Question:** Based on the visual characteristics of the two feline varieties depicted in the images, specifically the spotted coat of the first and the distinct ear structure of the second, what hybrid breed results from their cross? regarding the breed identified in the first part, what is the stated ultimate aim of its development according to the text?

A: The ultimate aim is to produce a cat that looks similar to a serval by introducing recent serval blood and then breeding back to domestic lines.",
B: The ultimate aim is to produce a cat with a leopard-like spotted coat that is smaller and more manageable than a Bengal, with no focus on serval resemblance.,
C: The ultimate aim is to standardize an extreme Oriental-type head and ear set on a spotted pattern, prioritizing show-ring silhouette over resemblance to any particular wild species.,
D: The ultimate aim is to produce a cat that looks similar to a serval, without using any recent wild cat blood.

Answer: D

Ground Truth Text Cue: Serengeti

## Reasoning with Textual cue

**text_cue_generated_gpt:**
The first cat has a golden, leopard-like rosetted/spotty coat (Bengal), and the second shows prominent outward-curled ears (American Curl); crossing them yields a Bengal Curl hybrid. The Bengal's stated ultimate development aim is to create a domestic cat with an exotic "wild" spotted look but a gentle pet temperament
.

**text_cue_generated_gemini:**
The images depict a Bengal cat, identified by its leopard-like rosetted coat, and an Oriental Shorthair, distinguished by its large, triangular ears. Crossing these two breeds produces the Serengeti cat, a hybrid developed with the ultimate aim of resembling the wild Serval without incorporating wild bloodlines.

## Reasoning with Visual cue

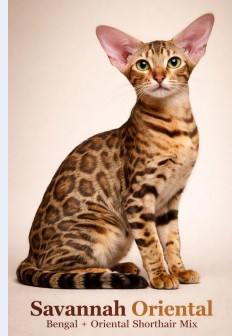
GPT Generated Visual Cue

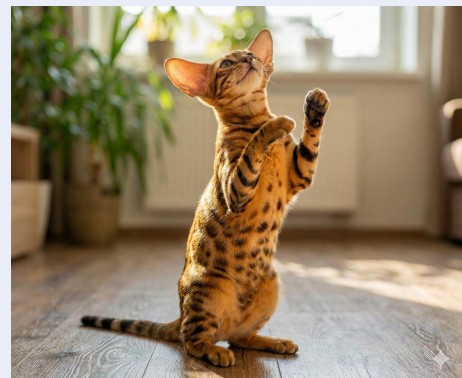
Gemini Generated Visual Cue

*Figure B.4.* **Case 4: Hybridisation reasoning (attribute composition).** The question couples fine-grained visual attributes with a text-described goal, so the intermediate cue must preserve discriminative appearance cues while aligning them with the referenced description.

## Inpainting reasoning question in UFO

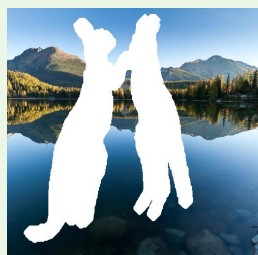

Input Images

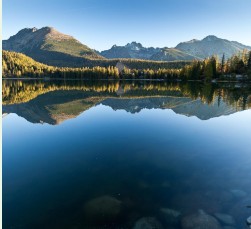

GT Image cue

**Question:** After restoring the occluded center section to reveal the full structure on the shoreline, what 2D geometric shape best describes the front profile of this building?

A: Triangle,
B: Parallelogram,
C: Trapezoid,
D: Diamond

Answer: A

Ground Truth Text Cue: The large building located at the center of the shoreline, nestled in the trees.

## Reasoning with Textual cue

**text_cue_generated_gpt:**
A shoreline building appears symmetrical; when the occluded middle is restored, its front face forms a broad "A" silhouette with two slanted sides meeting at a peak, like a triangular gable. The overall front profile is best described as a triangle.

**text_cue_generated_gemini:**
The building on the shoreline features a steeply pitched roof with sides that slope upward to converge at a sharp central apex, creating a distinctive triangular profile that is clearly mirrored in the lake's reflection.

## Reasoning with Visual cue

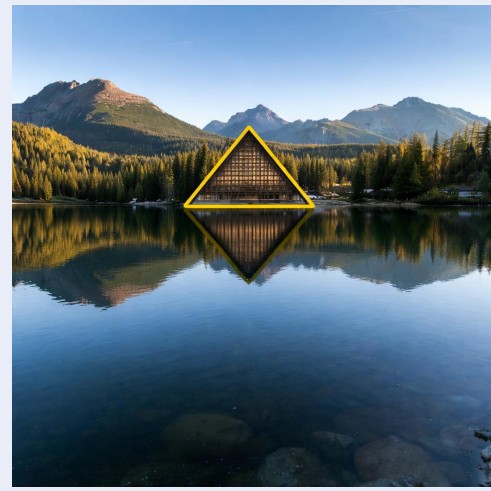

GPT Generated Visual Cue

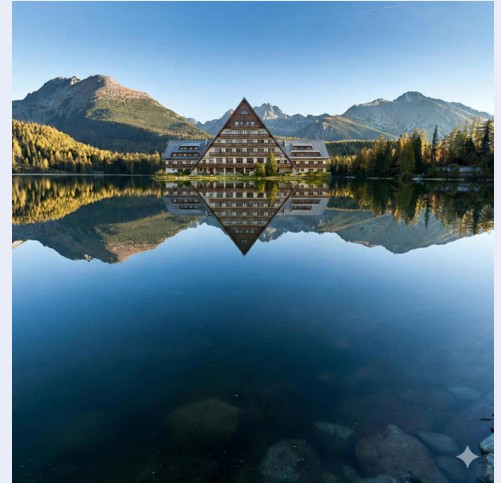

Gemini Generated Visual Cue

*Figure B.5.* **Case 5: Inpainting-based reasoning (reconstruction).** Correct answering depends on recovering the occluded structure and summarizing its salient geometric profile in the cue, revealing whether reconstruction contributes causally to the final decision.

## Jigsaw reasoning question in UFO

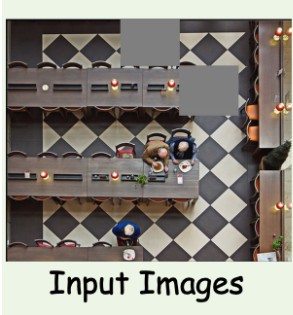

**Input Images**

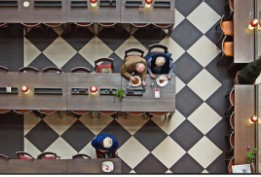

**GT Image cue**

**Question:** Which option correctly fills the missing areas, determining from the reference whether the concealed table section contains a plant, a person, or neither?,

A: 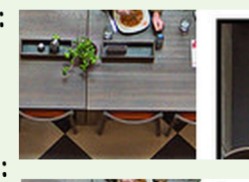 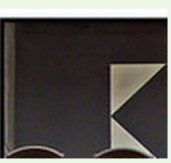   B: 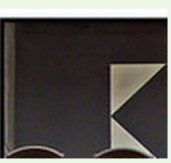

C: 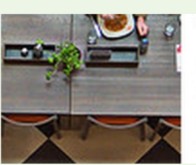 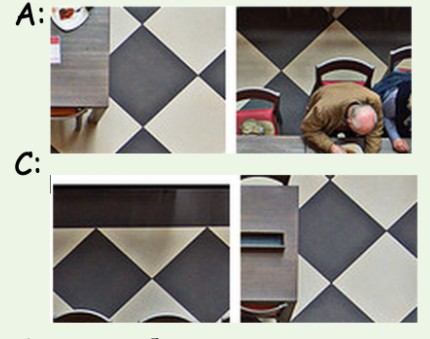   D: 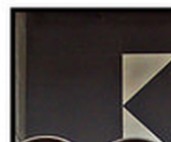

Answer: C

Ground Truth Text Cue: The unmasked top row has no plants or people; select the option showing the empty table and top wall edge

## Reasoning with Textual cue

**text_cue_generated_gpt:**
The building on the shoreline features a steeply pitched roof with sides that slope upward to converge at a sharp central apex, creating a distinctive triangular profile that is clearly mirrored in the lake's reflection.

**text_cue_generated_gemini:**
Based on the visual evidence in the reference image, here is the textual cue: Observe the vertical column of tables in the center of the layout; since the visible tables in the rows below each feature a small plant, the concealed table section in the top row should follow this pattern and also contain a plant.

## Reasoning with Visual cue

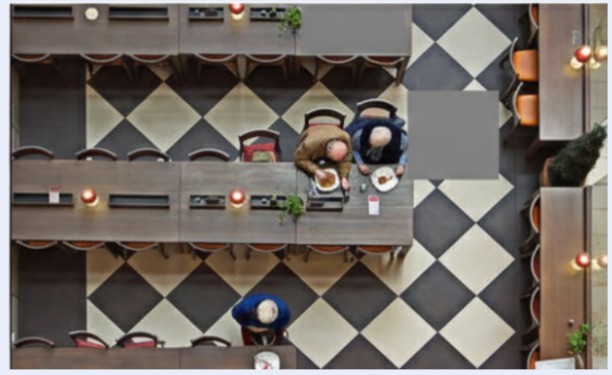

GPT Generated Visual Cue          Gemini Generated Visual Cue

*Figure B.6.* **Case 6: Jigsaw reasoning (compositional completion).** The task requires selecting the correct missing patch by matching local visual patterns, where the cue must encode the specific spatial correspondence needed for the completion.

## Logical reasoning question in UFO

Graph Representation of Vertices and Edges

Input Images

Graph Representation of Vertices and Edges

GT Image cue

**Question:** After the operation Remove the single line segment connecting the central node v7 to the right mid-level node v3. Calculate the ratio of the sum of degrees of all vertices with a degree strictly greater than 2, to the total number of edges in the graph.

A: 10/9,                    B: 3/2,
C: 4/3,                     D: 5/3

Answer: C

Ground Truth Text Cue: Remove the single line segment connecting the central node v7 to the right mid-level node v3.

## Reasoning with Textual cue

**text_cue_generated_gpt:**
Graph has 7 labeled vertices with a central node v7 connected to all six outer nodes; additionally, outer nodes form a ring with edges v1–v2–v3–v4–v5–v6–v1. Remove edge v7–v3; then degrees are v7=5, v3=2, others=3, with total edges 11.

**text_cue_generated_gemini:**
The graph consists of a hexagonal cycle (v1-v2-v3-v4-v5-v6-v1) with a central vertex v7 connected to vertices v3, v4, v5, and v6. This topology results in a total of 10 edges, where v7 initially has a degree of 4, vertices v3 through v6 have a degree of 3, and vertices v1 and v2 have a degree of 2.

## Reasoning with Visual cue

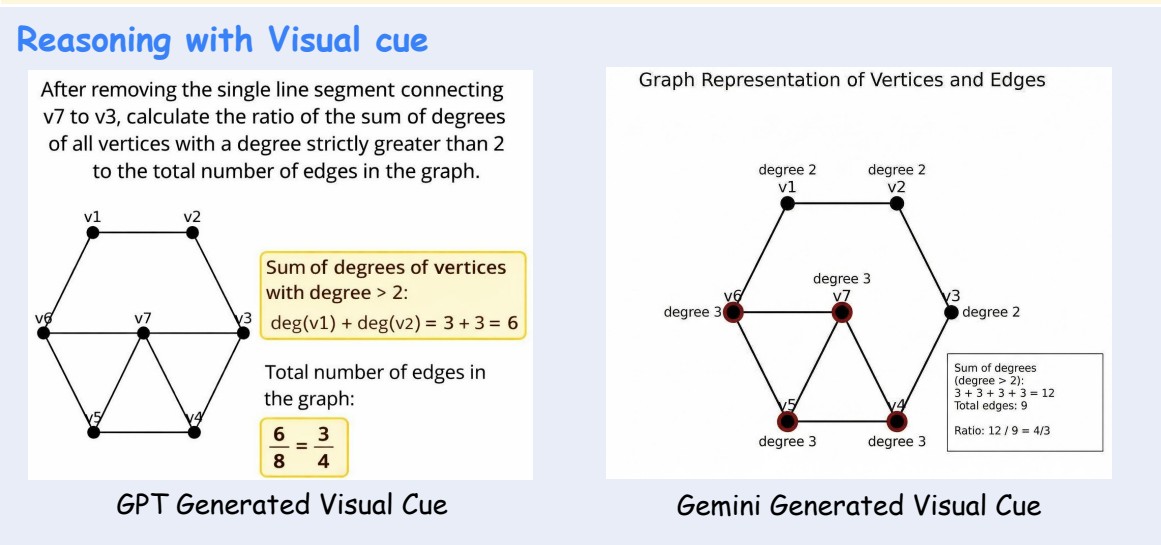

GPT Generated Visual Cue          Gemini Generated Visual Cue

*Figure B.7.* **Case 7: Logical graph reasoning (state transition via edit).** The cue formalizes the intervention (edge removal) and its structural consequences, enabling the answer step to depend on updated degrees/edges rather than a shortcut guess.

## MultiTbale reasoning question in UFO

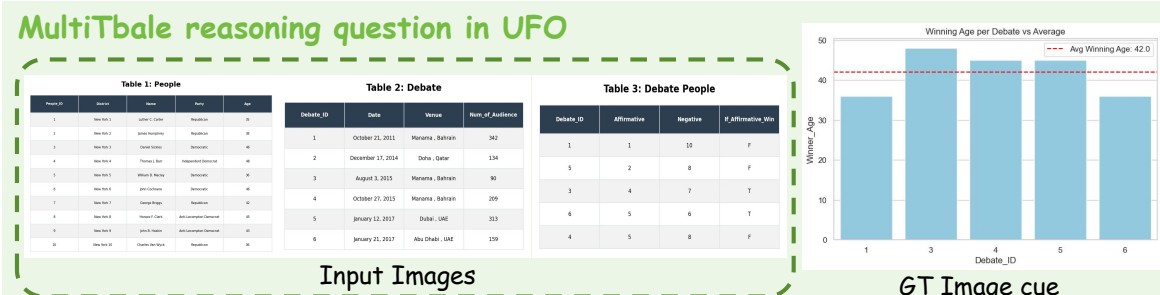

Input Images — GT Image cue

**Question:** Compute the average age of the winning speaker for all debates with recorded outcomes. Identify the dates of the debates where the Negative side won and the Negative speaker's age was strictly greater than this calculated average winning age.

A: January 12, 2017 and October 27, 2015

B: August 3, 2015 and January 12, 2017,

C: October 27, 2015,

D:December 17, 2014 and October 27, 2015

Answer: A

Ground Truth Text Cue: Merge tables to associate speaker ages, calculate the average winner's age (42.0), and retrieve the dates for debates where the Negative winner was strictly older than that average.

## Reasoning with Textual cue

**text_cue_generated_gpt:**
Locate debates with a recorded winner; for each, use the winning side's speaker age to compute the overall average winning age. Then filter rows where winner = "Negative" and Negative speaker age > that average, and report the corresponding debate dates.

**text_cue_generated_gemini:**
Cross-reference the Debate People and People tables to determine the ages of winning speakers based on the `If_Affirmative_Win` column ('T' for Affirmative, 'F' for Negative) and calculate the average. Then, identify dates from the Debate table for debates where the Negative side won and the speaker's age is strictly greater than this average.

## Reasoning with Visual cue

**Average Age of Winning Speaker:** 38.67

Dates of debates where the Negative **side won**, and the Negative speaker's age was strictly greater than 38.67:

| Date | Speaker as Negative | Age |
|------|---------------------|-----|
| ☑ Oct 21, 2011 | Charles Van Wyck | 36 |
| ☑ Jan 12, 2017 | George Briggs | 42 |

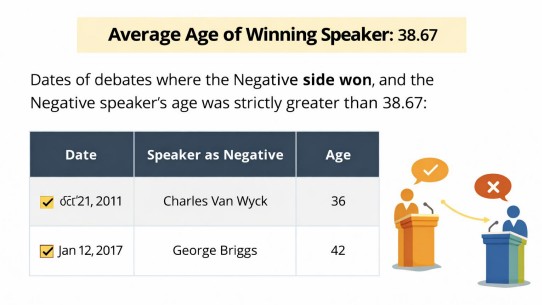

GPT Generated Visual Cue

| | | | Table 4: Winning Speakers, Ages, and Target Debate Analysis | | |
|---|---|---|---|---|---|
| Debate_ID | Date | Winning_S Side | Winning_Sp-eaker_Name | Winning_S peaker_Age | Negative_Win_&_A ge_Above_Average |
| 1 | October 21, 2011 | Negative | Charles Van Wyck | 36 | |
| **5** | **January 12, 2017** | Negative | Horace F. Clark | **45** | **Yes (45 > 42)** |
| 3 | August 3, 2015 | Affirmative | Thomas J. Barr | 48 | |
| 6 | January 21, 2017 | Affirmative | William B. Maclay | **36** | |
| 4 | October 27, 2011 | Negative | Horace F. Clark | **45** | **Yes (45 > 42)** |

Average age of all winning speakers: 42 years. Target debates (Negative win, age > 42) are highlighted in the last column.

Gemini Generated Visual Cue

*Figure B.8.* **Case 8: Multi-table reasoning (relational composition).** The cue must specify the join keys and the derived statistic before filtering, making the evidence chain explicit and auditable across intermediate operations.

## Multi-view reasoning question in UFO

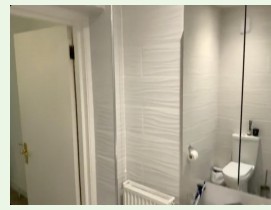

Input Images

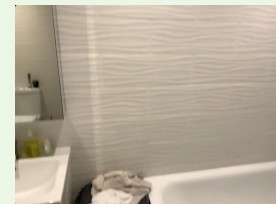

GT Image cue

**Question:** The second image shows the result of panning the camera to the right (clockwise) from the starting view. What large plumbing fixture is revealed in this new view?

A: Shower Stall,
B: Toilet,
C: Bathtub,
D: Vanity with integrated sink basin

Answer: C

Ground Truth Text Cue: Identification relies on matching the distinctive horizontal wavy texture pattern on the white wall tiles to orient the views.

## Reasoning with Textual cue

**text_cue_generated_gpt:**
After panning the camera to the right from the initial view, a large white bathtub is revealed along the bathroom wall, occupying a prominent portion of the new frame.

**text_cue_generated_gemini:**
Based on the provided image, here is the textual cue: The image displays a bathroom interior featuring a large mirror on the right wall that clearly reflects a white toilet and toilet roll holder located on the opposite side of the room. A white radiator is also visible, mounted on the textured tiled wall to the left of the mirror.,

## Reasoning with Visual cue

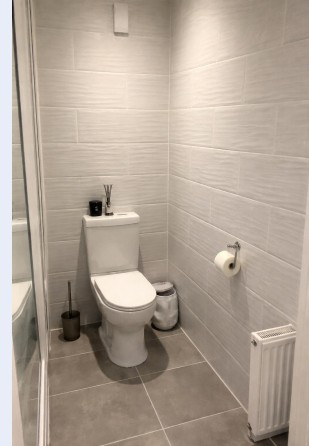

GPT Generated Visual Cue

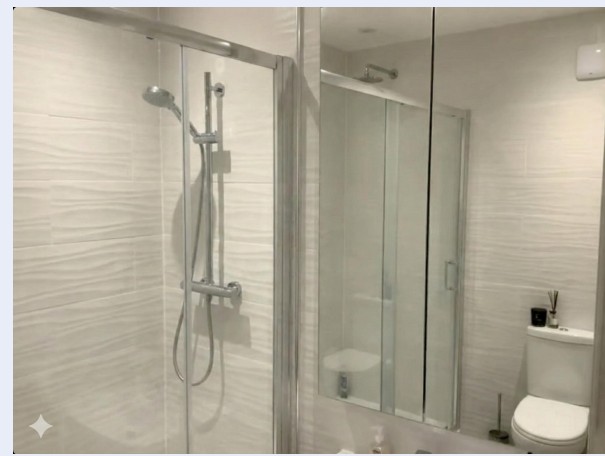

Gemini Generated Visual Cue

*Figure B.9.* **Case 9: Multi-view reasoning (view alignment).** The instance tests whether the cue captures the correspondence between views via shared textures or landmarks, supporting a correct identification after viewpoint change.

## Physics reasoning question in UFO

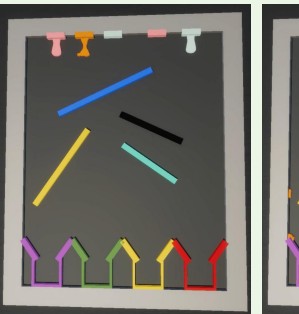 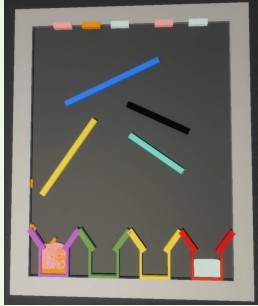

Input Images      GT Image cue

**Question:** Which container catches the liquid falling from the orange emitter?

A: second container from the left (the purple-green adjacent pocket),
B: yellow container,
C: center green container,
D: purple container
Answer: D

Ground Truth Text Cue: The orange object rolls down the blue and yellow ramps to land in the purple container.

## Reasoning with Textual cue

**text_cue_generated_gpt:**
The liquid stream from the orange emitter falls directly into the container positioned immediately below it; identify the container aligned under the emitter's spout/outlet and intersecting the falling liquid path, not adjacent containers offset to the side.

**text_cue_generated_gemini:**
The orange emitter drops liquid onto a blue bar angled downwards to the left, which directs the flow onto a similarly angled yellow bar below. This path guides the liquid into the leftmost purple container.

## Reasoning with Visual cue

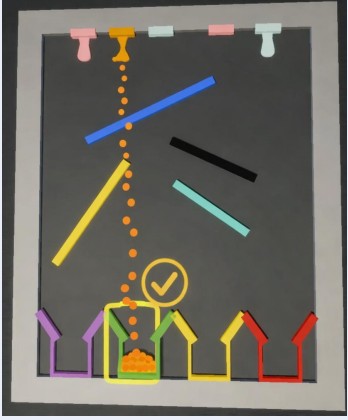
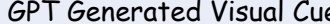

GPT Generated Visual Cue

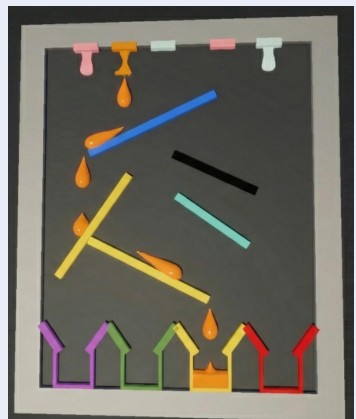

Gemini Generated Visual Cue

*Figure B.10.* **Case 10: Physics reasoning (causal trajectory).** The cue must encode the causal path of the liquid/object through intermediate ramps, so the final answer depends on a faithful physical trace rather than surface-level saliency.

## C. Limitations and Future Work

While UFO is designed to expose whether intermediate multimodal cues are causally utilized during reasoning, it does not aim to fully characterize all forms of multimodal intelligence. Our benchmark focuses on question-driven state transitions with explicitly defined future states, a formulation that enables controlled evaluation of evidential coupling but does not encompass open-ended reasoning scenarios in which the target state is implicit, evolving, or negotiated through interaction. Extending UFO to such settings would require mechanisms for tracking, validating, and attributing evidence across longer horizons, which we leave as an important direction for future work.

UFO currently targets vision–language reasoning, assuming that intermediate evidence can be expressed as textual and visual cues. In many real-world systems, however, state transitions unfold over richer and temporally continuous modalities, including audio, video, depth, or proprioceptive signals. Generalizing our evaluation protocol to these domains would require rethinking both the representation of intermediate cues and the notion of evidential sufficiency under temporal aggregation. We view such extensions as complementary to the current benchmark and defer their exploration to future studies.

## D. Ethical Considerations

UFO is an evaluation benchmark and does not introduce new generative capabilities or deployment mechanisms. As such, it does not directly amplify risks associated with content generation, misinformation, or model misuse. Nevertheless, by promoting evaluation protocols that emphasize interpretable and intervenable intermediate representations, UFO may indirectly support the development of more controllable and accountable multimodal systems.

At the same time, evidential coupling alone does not guarantee correctness, fairness, or robustness. A model may rely consistently on its intermediate cues while those cues reflect biased or incomplete interpretations of the input. We therefore view UFO as complementary to existing efforts in safety, bias mitigation, and robustness evaluation. Ensuring that intermediate evidence is not only causally effective but also socially and ethically grounded remains an open challenge beyond the scope of this work.

