# OpenReview forum: "Do Vision and Text Cues Exhibit Evidential Coupling? UFO: A Benchmark for Compositional Multimodal Reasoning in Unified Models"
_ICML.cc/2026/Conference — ICML 2026 regular_

### Official Review · Reviewer_AASV · 2026-02-26

**Soundness:** 2
**Presentation:** 2
**Significance:** 3
**Originality:** 3
**Overall Recommendation:** 3
**Confidence:** 3

**Summary:**

The paper studies the coupling between evidence modalities in multi-modal LLMs. The task: the model is given a question and an image. Then, the model needs to generate intermediate evidence (either an image, text, or both) that helps answer the question. Finally, the question is answered based on this evidence. The goal is to study how the modality of the intermediate evidence impacts the quality of the final answer.

To measure this, the paper proposes a new benchmark that involves three categories of tasks: 1) State determination: all the information required is provided in the input question and image 2) state reconstruction: some of the information is omitted from the input (e.g., a patch in the image is masked) and the goal is to reconstruct the input and answer a question about it (e.g., what is the shape of the object in the area of interest after reconstruction?) and 3) state augmentation: while all the information is provided in the input, there is additional logic or information needed to solve the problem (e.g., the image represents a geometry problem but additional information about how to solve the problem or intermediate steps are needed to generate the final answer).

Based on the experiments using this benchmark, the authors conclude that there is a significant modality gap in current multi-modal LLMs, and across the three categories, the evidence modalities are not consistently coupled together. The authors also introduce a checklist-based evaluation strategy powered by LLM-as-a-judge to better understand the quality of the generated evidence from different aspects.

**Compliance With Llm Reviewing Policy:**

Affirmed.

**Final Justification:**

Although the authors provided additional details during the rebuttal period, the answers are not satisfactory or sufficient. Specifically:

- The expected behavior (evidential coupling across modalities) is not well justified with the current definition provided by the paper. Please see my response to the authors in the Rebuttal Acknowledgment.
- The paper lacks important details about data construction. While authors briefly discuss these details during the rebuttal, I believe that, as a core contribution of the paper, data construction and evaluation protocol should be discussed in the paper in detail beyond brief discussions during the rebuttal.
- As acknowledged by the authors, the analytical experiments using LLM evaluators are not highly reliable. I understand the limitations of LLM evaluators. And while such experiments are acceptable as exploratory or preliminary understandings, it is difficult to accept them as main contributions without the required level of reliability.

Therefore, I maintain my initial score.

**Key Questions For Authors:**

Please refer to the Strengths and Weaknesses section.

**Limitations:**

yes

**Strengths And Weaknesses:**

## Strengths

1. The paper studies an interesting problem with increasing importance in the future. As vision-language models are more frequently used to take action in the real world, it is important to have a better understanding of the reasoning process that guides their decision-making.
2. The paper experiments on a large number of proprietary and open source models with different architectures which makes the conclusions applicable to a wider range of audience.
3. The checklist based evaluation is a step in the right direction as it provides more fine-grained and actionable findings compared to relying on final performance alone.

## Weaknesses

1. As the core finding of the paper, the authors claim that evidence across modalities is not coupled together since providing evidence in each modality alone improves the performance but providing evidence from both modalities does not improve the performance further (Line 370). This is a very reasonable behavior since, depending on the task, evidence in one modality can be more effective and even sufficient for answering the question. For example, for the question “What is the offspring that results from interbreeding the animals in the left and right images called?”, text evidence that just mentions the name of the breed in each image is sufficient for answering the question with no benefits for additional model-generated visual clues. So, it is not clear how this observation supports the claim and more importantly, is it reasonable to expect clues in both modalities to be helpful for each question?

2. Missing implementation details about the answer generation step. The paper provides the prompts for the evidence generation steps. However, the prompts for generating the final answer are not provided. Moreover, when generating the final answer based on the evidence, is the original image also used as part of the input, or is the answer generated solely based on the evidence? This is a critical detail for interpreting the results.

3. The details of how the dataset is constructed are not clear. Only the process for one task type is discussed and the description is very high level, and thus is not possible to assess the quality of the data. For example, Section 3.2 (stage 1) mentions that LLMs are used to generate QA pairs. What is the prompt used for generating these QA pairs? How is Gemini used to cross check GPT-5.1 output?

4. For the checklist evaluation process, Table 6 reports that agreement between humans and the LLM judge is at most 64% on average, which is not sufficient for driving reliable conclusions from the evaluation results.

5. While the contributions are easy to understand, the writing is not clear and requires reading the paper multiple times to understand the core contributions and the problems addressed.



- **Minor Comment:** There might be a mistake in Figures C.2 and C.6 where the evidence or the answer does not match the question.
- **Optional Suggestion:** I might be missing some details. But, I think framing the problem in terms of task types makes it easier to understand than the current world-state framing. Also, the paper switches between discussing connection between the two modalities and connection between generation and understanding tasks. A clear connection between the two paradigms or better separation might make the paper an easier read.

---

> ### Author Rebuttal · Authors · 2026-03-31
>
> We thank the reviewer for recognizing our systematic study of cross-modal evidence and our process-level evaluation, which offers a more interpretable and actionable assessment beyond final-answer accuracy.
>
> > ### Why Evidential Coupling Matters？
>
> 1. We agree that not every instance requires both modalities. Our claim is therefore not that joint cues must universally outperform single-modality cues.
>
> 2. Our contribution is to evaluate a different property: if textual and visual cues are generated as evidence, they should be mutually consistent, composable, and jointly usable under a fixed answer protocol. From a design perspective, if one modality alone is sufficient, additional cues should not introduce negative effects, at least when UFMs are well designed, as cues across modalities should be aligned and consistent, and models should adapt to cross-modal information.
>
> 3. UFO is designed to test exactly this property. Because the original image is retained at answer time, the benchmark measures whether self-generated cue(s) provide incremental coordinated evidential value, rather than whether every question intrinsically needs both modalities.
>
> 4. Empirically, single-modality cues can help, but joint cues often fail to provide stable additional value and can even hurt performance. This does not mean both modalities are always necessary; it shows that current UFMs often fail to turn self-generated multimodal outputs into reliably usable joint evidence.
>
> 5. This is the key contribution of UFO: it diagnoses whether multimodal generation becomes causally useful evidence, rather than merely additional multimodal content.
>
> > ### Answer Stage Clarification
>
> 1. In the final answer stage, the model always receives Question + original image + generated cue(s). The original image is never removed.
>
> 2. Accordingly, UFO measures the incremental evidential value of generated cues beyond the original perceptual input, rather than replacing that input with generated evidence.
>
> 3. We will make the answer stage template explicit in the revision:
> Question: [question], optional Options:, optional Reference Text Cue: [cue], generated visual cue when enabled, followed by “Please only provide a short and clear answer. Do NOT output your reasoning process.” and “Answer:”.
>
> > ### Dataset Construction Transparency
>
> 1. We agree that the current paper under-specifies the construction pipeline. We will make it explicit as a two-stage generation + dual-model validation + human verification process.
>
> 2. For generation, GPT first produces a cue-inducing question from the input image and task-specific requirements. It then generates a second variant conditioned on the corresponding ground-truth cue(s), so that the final question preserves the intended cue-dependent causal relation.
>
> 3. For validation, GPT and Gemini act as independent checkers. Each is given the same tuple (question, ground-truth cue(s), answer) and evaluates visual necessity, cue relevance, and answer consistency. An item is retained only when both judges accept it.
>
> 4. Human annotators then verify accepted items and correct cases with minor errors.
>
> 5. We will release the exact QA generation prompts, validation prompts, and acceptance criteria in the revision. This unified generation and filtering protocol is itself an important contribution of UFO.
>
> > ### LLM Judge Reliability
> 1. Our main conclusion does not rely on the LLM judge. Weak evidential coupling is already supported by the answer-matching results in Table 3, which are independent of checklist judging.
>
> 2. To verify that this is not an artifact of judge noise, we additionally analysed approximately ~8,000 human-labeled samples covering both cues and answers from GPT-5.1 and Gemini-3-Pro. The same directional conclusion holds.
>
> 3. Therefore, the process-level analysis should be interpreted as fine-grained diagnosis.
>
> 4. We also systematically study judge models and protocols (Tables 6–7), improving agreement from 64% in Table 6 to 67.4% overall and 72.8% on visual cues in Table 7.
>
> > ### Figures and Writing Clarity
> 1. We confirm minor presentation errors in Figures C.2 and C.6, the revised version is in https://anonymous.4open.science/r/ICML_rebuttal-6B85/README.md. These are display issues only and do not affect any results.
> 2. We have corrected them and will update the appendix in the revision.
> 3. We will also streamline the writing to make the setup and contributions more immediately clear.
>
> > ### Paradigm Framing and Readability
> 1. We agree that organising UFO by task types provides a clearer reader-facing structure. In the revision, we will foreground task types and use the world-state view only as a unifying abstraction.
> 2. We will also explicitly distinguish generation and understanding coupling from cross-modal evidential coupling, so the paper does not conflate capability with evidence consistency.

---

> > ### Author Rebuttal · Reviewer_AASV · 2026-04-04
> >
> > Thank you for your detailed response.
> >
> > 1.
> > In their answer, the authors mention the following:
> >
> > `Our contribution is to evaluate a different property: if textual and visual cues are generated as evidence, they should be mutually consistent, composable, and jointly usable under a fixed answer protocol. From a design perspective, if one modality alone is sufficient, additional cues should not introduce negative effects,at least when UFMs are well designed, as cues across modalities should be aligned and consistent, and models should adapt to cross-modal information.`
> >
> > However, they do not explain why this is the desired behavior. It is not difficult to think of examples where providing evidence in one modality is more suitable than the other modality. And if both are provided to the model, the information in the non-desired modality could simply act as noise or sub-optimal information for this question and degrade performance.
> > For example, for a math question, an intermediate formula in text format is helpful but representing this formula with shapes and figures as an image is not the best approach and it is not surprising or unexpected if it degrades performance.
> >
> > 2.
> > I appreciate the additional details regarding the data construction process. As the authors mentioned, while the data construction process is an important contribution of their work, it is underspecified in the main paper.
> > I do not think it is possible to carefully evaluate a major component of the paper with a brief description during the rebuttal period.
> >
> > 3.
> > Regarding LLM reliability, I understand this analysis is separate from the main results. But it is still claimed as a major contribution of the work, and 67% agreement with human judgment is not reliable enough to justify the conclusions.

---

> > > ### Author Response · Authors · 2026-04-06
> > >
> > > > ## UFO Evaluates Evidential Coordination.
> > > 1. The same underlying evidence may vary in its suitability across modalities. In UFO, we ensure that cues can be expressed naturally in both modalities. For example, in mathematical tasks, we do not include formulas as intermediate cues. Instead, we include only auxiliary lines as cues, which can be described in text and represented visually in images.
> > > 2. One core vision and motivation of UFMs is that cross-modal generative and understanding capabilities can benefit from each other. Therefore, in the cue generation stage, UFO is designed to evaluate the consistency and alignment of cues across modalities. Additionally, in the answering stage, UFO evaluates the utilisation of cross-modal cues in reasoning. Well-designed UFMs should generate aligned cues and effectively utilise them for question answering. Cues may have different levels of suitability across modalities. However, well-designed UFMs should be able to recognise this, and additional cues from different modalities should provide complementary information rather than introduce inconsistency. As these cues are not arbitrary external hints, the model is encouraged to generate useful cues and produce answers in a unified manner.
> > > 3. Accordingly, the UFO protocol directly evaluates whether self-generated cues remain aligned and coordinated once they are produced and reused as evidence. Differences in modality suitability may affect the extent to which a cue is helpful, but the benchmark assesses whether these cues remain aligned with the same underlying state rather than collapsing into inconsistency or noise when used jointly.
> > > 4. Existing UFM benchmarks largely separate modalities, which limits their ability to evaluate whether generated cues are actually reused as evidence in reasoning. In contrast, UFO evaluates evidential coordination: whether self-generated multimodal cues can function as grounded, state-consistent, and jointly usable evidence for future-state reasoning under a unified answer protocol.
> > >
> > > > ## UFO Uses a Unified Construction Pipeline.
> > > 1. We agree that the construction pipeline should be described more explicitly in the current draft. However, the paper does not present an isolated task-specific procedure. Section~3.2 already outlines a shared curation backbone, including image collection, QA generation, followed by LLM-assisted and human verification.
> > > 2. The paper explicitly presents the elaborated case as the most complex one and uses it as a representative example of this shared pipeline. The remaining task types follow the same general process of generation, validation, and verification, with task-specific adjustments to cue construction.
> > > 3. This unified pipeline is a core contribution of UFO. The benchmark is not a collection of unrelated tasks, but a single evidential framework that organises diverse task types under shared requirements for cue dependency, evidence use, answerability, and validation.
> > >
> > > > ## UFO Adds Process-Level Diagnosis Beyond MCQ Results.
> > > 1. We agree that the reported human--LLM agreement is not high enough for these judgments to serve as a stand-alone basis for strong conclusions. However, this does not affect the paper's main findings, because our central empirical results are established by MCQ answer performance under fixed protocols (Table~3) and do not rely on LLM-based judgments.
> > > 2.The role of this process-level analysis is therefore diagnostic rather than definitive. Its purpose is to move beyond endpoint accuracy and reveal where generated evidence succeeds or fails, e.g., in alignment, relevance, completeness, and consistency. It is a secondary layer for interpreting the phenomenon, not a replacement for the primary answer-based evaluation. Without this layer, evaluation would stop at whether the final answer is correct; with it, UFO can additionally analyze whether the generated evidence itself is aligned, relevant, and internally consistent.
> > > 3. Our contribution here is not to present LLM-based judgments as a perfect adjudicator, but to introduce a structured process-level diagnostic framework for multimodal evidence evaluation built on top of the MCQ-based findings. More importantly, this limitation is not unique to UFO: reliability, bias, and human alignment remain open challenges for LLM-based evaluation more broadly, which is precisely why we do not use this component as the evidentiary foundation of the paper. Compared with other research that uses LLMs as judges, UFO achieves comparable performance, for example, on MMRB2 [1]. We believe our systematic exploration provides valuable insights for subsequent research.
> > > 4.The process-level analysis is used only to characterise directional trends and failure modes. These trends are further cross-checked through additional human-labelled analysis rather than treated as an independent proof mechanism.
> > >
> > >
> > > [1] Multimodal RewardBench 2: Evaluating Omni Reward Models for Interleaved Text and Image, arxiv 2025.

---

### Official Review · Reviewer_ZiK6 · 2026-03-09

**Soundness:** 3
**Presentation:** 2
**Significance:** 3
**Originality:** 3
**Overall Recommendation:** 4
**Confidence:** 4

**Summary:**

This paper introduces UFO, a benchmark designed to evaluate compositional multimodal reasoning in Unified Foundation Models (UFMs) by testing the evidential coupling between generated visual and textual cues during reasoning. UFO organizes tasks into three state-transition categories: state determination, state reconstruction, and state augmentation, and evaluates models on their ability to generate intermediate multimodal cues and use them to answer questions about future states. Experimental results reveal that while models can generate plausible cues, they often fail to exhibit strong evidential coupling, relying on textual shortcuts rather than robust cross-modal grounding. The paper highlights a gap in current UFMs' ability to consistently use multimodal evidence, suggesting that the coupling of reasoning and generation is still a non-trivial challenge. UFO serves as a diagnostic tool to explore and evaluate multimodal reasoning behaviors in future models.

**Compliance With Llm Reviewing Policy:**

Affirmed.

**Final Justification:**

Thanks for the author's detailed response, which has addressed most of my concerns. I will maintain my positive score and increase my confidence.

**Key Questions For Authors:**

1. Why is it necessary to solve these tasks with a unified model rather than directly relying on multimodal language models? Tasks such as Multi-view, Geometry, and Jigsaw in the UFO benchmark have already appeared repeatedly in traditional MLLMs evaluations, where they have performed quite well.
2. Could the authors provide some performance results of MLLMs on UFO to further demonstrate the necessity of the unified model joint reasoning approach?
3. I noticed that the examples provided in the appendix are all failures. Could the authors provide some successful examples to demonstrate the advantages of joint reasoning?

**Limitations:**

See above

**Strengths And Weaknesses:**

+ **Strengths:**
  + The paper evaluates 12 unified multimodal models, covering proprietary and open-source systems across different architectures.
  + Beyond final accuracy, the study evaluates cue quality along several dimensions (e.g., relevance, causal utility, specificity), enabling deeper diagnostic insights into model reasoning behavior.
  + The figures illustrating the reasoning pipeline and benchmark composition effectively clarify how UFO operates. Additionally, the authors provide numerous examples in the appendix, which contribute to a better understanding of the benchmark.

+ **Weakness:**
  + Figure C.2, Case 2 seems to contain an error. The question asks the participant to choose between using the left hand, the right hand, or both hands simultaneously, yet the options provided list four different actions, which is confusing.
  + The analysis of the experimental results is not sufficiently in-depth. In Section 4.3, the authors simply note that visual cues are helpful in some cases but not in others, leaving a gap in the analysis. While Section 4.4 addresses the negative impact of image quality, the comparison between Table 3 and Table 5 suggests that image quality and task performance are not strongly correlated. For example, the UniWorld-V1 model shows a relatively high joint performance of 27.28 on State Determination, yet both the textual and visual quality in Table 5 are quite low. I think the current analysis of the experimental results is somewhat shallow and does not fully explain the model's performance in the main results.
  + I believe the paper lacks a clear justification for the necessity of the joint reasoning paradigm. Some tasks in the benchmark, such as Jigsaw and Multi-view, do not necessarily require the generation of intermediate images to be solved. As a result, traditional MLLMs with purely text-based reasoning can perform well. The need to enforce the use of a Unified model for Evidential Coupling has not been adequately demonstrated.

---

> ### Author Rebuttal · Authors · 2026-03-31
>
> We thank the reviewer for recognizing our comprehensive evaluation across diverse models and our multi-dimensional analysis, which reveals key gaps between cue generation and actual evidential use.
>
> > ### Why Endpoint Accuracy and Cue Quality Can Diverge
> 1. The mismatch between Table 3 and Table 5 is acceptable in UFO and is not anomalous, because the original image is retained at answer time, allowing endpoint accuracy to remain non-trivial even when self-generated cues are weak. From a design perspective, when UFMs are well designed, models should generate cross-modal consistent cues and make effective use of evidential cues, leading to incremental performance gains compared with Direct.
> 2. Therefore, individual endpoint performance is not a direct proxy for evidential coupling, as it depends on cue quality, consistency between cues, and the effectiveness of cue utilisation. Table 3 evaluates whether the model can reach the correct answer under a cue-conditioned protocol, while Table 5 assesses the quality of the self-generated evidence.
> 3. The UniWorld-V1 example is not an exception; it reflects the phenomenon that UFO is designed to measure, where models can achieve non-trivial joint accuracy despite low-quality cues, suggesting that cross-modal cues jointly provide complementary information and yield performance gains over the individual use of textual or visual cues.
> 4. This is a core contribution of UFO, as it disentangles solving the task from solving it with causally useful self-generated evidence.
>
> > ### Why Unified Models Are Necessary.
> 1. We agree that many underlying tasks in UFO, such as Jigsaw, Multi-view, and Geometry, can be answered by strong MLLMs. However, answerability is not the target of UFO.
> 2. Our contribution is to evaluate evidential coupling, that is, whether a model can generate intermediate multimodal cues, use them in answering, and be audited on whether those cues constitute causally useful evidence. Standard MLLM evaluations typically stop at final-answer correctness and therefore do not test this property.
> 3. Accordingly, UFO does not claim that these tasks are impossible for MLLMs. Rather, it introduces a new generate–consume–audit protocol that distinguishes solving the task from solving it with grounded self-generated multimodal evidence.
> 4. This is why unified-model-style evaluation is necessary here. The benchmark is designed to assess whether self-generated multimodal outputs become usable joint evidence, not merely whether the final answer is correct. The table further shows that text-only gains are limited and inconsistent, reinforcing that answerability alone is not evidential coupling.
>
> ---
> | Model             | Method   | Avg.  |
> |-------------------|----------|-------|
> | Internvl-3.5-8b   | Direct   | 38.28 |
> |                   | Text cue | 40.17 |
> | Qwen3-VL-30b      | Direct   | 43.73 |
> |                   | Text cue | 45.49 |
> | Qwen3-VL-8b       | Direct   | 42.92 |
> |                   | Text cue | 39.96 |
> ---
>
> > ### MLLM Baseline Results
> 1. We evaluated representative MLLMs on UFO under the same Direct and Text-Cue settings.
> 2. The results in the table confirm that some UFO instances are answerable by strong MLLMs. This calls for further contributions from UFMs.
> 3. However, this does not weaken our claim; it sharpens it. UFO is not only about whether a model can answer the question, but whether it can do so with causally useful self-generated multimodal evidence.
> 4. This contrast is exactly our contribution. Strong answerability does not imply evidential coupling, and UFO is designed to measure that gap directly.
>
> > ### Successful Cases and Appendix Examples
>
> 1. The appendix is not intended to suggest that joint reasoning only fails. Many examples are organised as contrastive cases to illustrate both benefits and failure modes.
> 2. Positive cases do exist. For example, Joint improves OmniGen2 on Reconstruction from 51.52 to 61.18, and GPT-5 on State Determination from 45.91 to 51.37 (Table 3). These improvements are consistent with Appendix B.1, where gains are strongest when visual grounding is essential.
> 3. We will add representative successful examples in the revision to better highlight our contribution. UFO diagnoses when joint reasoning yields causally useful evidence and when it does not.
> 4. In the revision, we will correct the illustration error in Figure C.2 and add representative examples, as shown at https://anonymous.4open.science/r/ICML_rebuttal-6B85/README.md
> . This further foregrounds our contribution, as UFO diagnoses when joint reasoning yields causally useful evidence and when it does not.

---

> > ### Author Rebuttal · Reviewer_ZiK6 · 2026-04-03
> >
> > Thanks for the author's detailed response, which has addressed most of my concerns. I will maintain my positive score and increase my confidence.

---

> > > ### Author Response · Authors · 2026-04-03
> > >
> > > Dear Reviewer ZiK6,
> > >
> > > Thank you very much for your kind feedback. We sincerely appreciate your time and effort throughout the review process.
> > >
> > > We are glad that our response helped further clarify the contribution and value of our work, and we are especially encouraged by your continued positive assessment and increased confidence in the paper.
> > >
> > > Best regards,
> > > The Authors

---

### Official Review · Reviewer_oN5J · 2026-03-10

**Soundness:** 3
**Presentation:** 3
**Significance:** 3
**Originality:** 4
**Overall Recommendation:** 5
**Confidence:** 4

**Summary:**

The authors propose UFO, a state-centric benchmark for UFMs, designed to evaluate two-step reasoning with explicit measures of evidential coupling based on cue consistency and causal contribution.
The authors conduct extensive experiments and provide detailed evaluations, offering a thorough analysis of the challenges that current UFMs face in multimodal reasoning.

**Compliance With Llm Reviewing Policy:**

Affirmed.

**Final Justification:**

Thanks for the rebuttal, most of my concerns are addressed. I have improved my score. It would be beneficial if the benchmark dataset were larger in scale.

**Key Questions For Authors:**

1.What is the proportion of data derived from each source dataset?

2.Do different models adopt different strategies when jointly utilizing textual and visual cues? If so, what are the specific mechanisms used by each model?

**Limitations:**

yes

**Strengths And Weaknesses:**

Strengths:

1. The authors identify three categories of challenges that may arise in multimodal reasoning, State Determination, State Reconstruction, and State Augmentation and construct corresponding questions by leveraging existing datasets and information from Wikipedia.

2. The paper categorizes complex multimodal reasoning into these three problem types and evaluates current UFMs accordingly. In addition, the authors design process-level evaluations for each category, enabling a more precise analysis of the specific difficulties UFMs encounter during multimodal reasoning.

3. The authors apply rigorous quality control procedures to construct a high-quality evaluation dataset and explicitly consider potential issues such as textual shortcuts and data contamination.

Weaknesses:

1. The dataset contains 3,936 questions in total; a larger overall scale would further strengthen the benchmark.

2. It would be helpful to report the proportion of questions derived from different source datasets.

3. The methodology for jointly applying textual and visual cues across different models could be explained in greater detail.

---

> ### Author Rebuttal · Authors · 2026-03-31
>
> We thank the reviewer for recognizing our structured decomposition into state transitions and our process-level evaluation design, which enables more fine-grained and diagnostic analysis of multimodal reasoning.
>
> > ### Scale and Coverage
> 1. We agree that a larger scale can further strengthen a benchmark. However, the main contribution of UFO is not raw size, but a new evaluation target and protocol for evidential coupling.
>
> 2. UFO already covers 3,936 questions, 10 task types, 3 state regimes, and both MCQ and open-ended settings, which are sufficient to support the paper’s protocol-level claims.
>
> 3. To our knowledge, UFO is the first benchmark for this capability. It is also already larger than the most closely related unified-model benchmarks (e.g., UniMMU, RealUnify, ROVER), which are typically below 2,000 instances. Thus, while further scaling is valuable, the current benchmark is sufficient to support the paper’s protocol-level contribution.
>
> > ### Source Composition
>
> 1. UFO does not inherit external QA annotations. All 3,936 questions are reconstructed by our team; external datasets provide only the raw images. This is a key part of our contribution. UFO is an independently constructed benchmark rather than a repackaging of existing QA datasets.
> 2. By image source, the question distribution is COCO 28.9%, Wikipedia/Web 23.2%, Ego-Exo4D 15.8%, MMMU 12.3%, Visual Jenga 12.1%, and ExpVid 7.7%.
> 3. We will add the full source composition table and per-stage filtering statistics in the revision to make benchmark provenance fully transparent.
>
> > ### How Textual and Visual Cues Are Jointly Used
>
> 1. There is no model-specific joint strategy in UFO. All models are evaluated under the same four schedules (Direct, Textual, Visual, Joint) with the same prompt format and decoding settings.
> 2. The answer-stage protocol is also fixed across models. Each model receives Question + original image + generated cue(s). This controls the external usage protocol and isolates the variable of interest, namely the incremental evidential value of self-generated cues.
> 3. What differs is only each model’s native internal mechanism. This is exactly our contribution. UFO keeps the protocol fixed so that differences in how models internally process and exploit multimodal cues can be compared meaningfully.

---

> > ### Author Rebuttal · Reviewer_oN5J · 2026-04-02
> >
> > Thanks for the rebuttal, most of my concerns are addressed. I would improve my score.

---

> > > ### Author Response · Authors · 2026-04-02
> > >
> > > Dear Reviewer oN5J:
> > >
> > > Thank you so much for the recognition of our responses. We are glad to receive your positive feedback!
> > > We are pleased to know that our rebuttal has fully addressed your concerns. We are deeply grateful for the professional guidance you provided throughout the review process.
> > >
> > > **Many thanks for your constructive comments, time, and patience.**
> > >
> > > Best regards and thanks,
> > >
> > > The Authors

---

### Official Review · Reviewer_Zvyj · 2026-03-12

**Soundness:** 3
**Presentation:** 2
**Significance:** 3
**Originality:** 3
**Overall Recommendation:** 4
**Confidence:** 2

**Summary:**

This paper presents UFO, a benchmark for evaluating Multimodal Reasoning in Unified Foundation Models (UFMs). Overall, this paper's primary contribution concerns the formalization and evaluation of "evidential coupling"—a mechanism designed to ensure that the images and text generated by unified models during intermediate reasoning steps serve as grounded, causal evidence for the final answer, rather than mere auxiliary output.
Overall, by requiring models to first generate explicit multimodal cues for future states (categorized as determination, reconstruction, and augmentation) and then condition their final answers on these cues, the benchmark forces models to demonstrate cross-modal consistency. This design effectively exposes models that rely on "textual shortcuts" rather than robust visual grounding.

**Compliance With Llm Reviewing Policy:**

Affirmed.

**Final Justification:**

Thanks for the authors' detailed responses, which successfully addressed my concerns. I have no further questions. Since my original rating is already positive, I'd like to maintain it.

**Key Questions For Authors:**

- In Table 5, how did the authors evaluate the process-level cue accuracy? To my knowledge, complex multimodal reasoning may involve multiple valid reasoning paths. How can we ensure the fairness of the evaluations?
- What are the evaluation costs of the proposed benchmark? To my understanding, the proposed UFO benchmark requires unifed models to generate both visual cues and textual cues, which may slow down the evaluation efficiency.
- Did the authors conduct any error case analyses? This may help better understand why joint cues failed to boost performance for some unified models.

**Limitations:**

yes

**Strengths And Weaknesses:**

Strengths:
- To the best of my knowledge, the framing of "evidential coupling" is a timely contribution. It moves beyond observational evaluation (does the model get the right answer?) to process-based evaluation (does the model get the right answer via the generated intermediate state?).

- The design of the proposed benchmark is rigorous. For instance, the use of "blind-modality audits" to filter out shortcuts is helpful. It ensures that the benchmark measures genuine perceptual reasoning rather than latent knowledge or language-only correlations.

Weakness:
- The paper writing is a little obscure, presenting obstacles for quickly understanding the *state* terms.

---

> ### Author Rebuttal · Authors · 2026-03-31
>
> We thank the reviewer for recognizing our formulation of evidential coupling and our shift to process-level evaluation, which provides a principled way to assess whether multimodal cues function as grounded evidence.
>
> > ### Fairness of Table 5
>
> 1. We do not evaluate cues by exactly matching a single reasoning path. Table 5 instead measures evidential sufficiency, that is, whether the cue provides the decision-critical evidence required to support the answer.
> 2. For each task, we identify its decision-critical factors (Appendix A.2.2) and mark each as verified, weak, missing, or contradicted. This makes the evaluation path-invariant, as different valid reasoning paths receive equal credit as long as they surface the necessary evidence.
> 3. We further restrict the rubric to verifiable evidence only, so cues that merely imply an answer without checkable support are not over-rewarded.
>
> > ### Efficiency and Practicality
>
> 1. As a benchmark, UFO is designed for diagnostic value, not minimal inference cost. Its goal is to expose schedule effects, cue quality, and evidence quality that endpoint-only QA benchmarks cannot reveal.
>
> 2. That said, the overhead is quantifiable and modular. On H100 80 GB (bfloat16), Direct is the 1× baseline (∼2 s per sample), Textual is ∼2×, Visual is ∼2–5×, and Joint is ∼4–6×, depending on the generator.
>
> 3. Importantly, UFO does not require every study to use the full Joint setting. Researchers can use Direct for answerability, Textual for low-cost diagnosis, and Joint when full evidential coupling analysis is needed.
>
> > ### Error Analysis and Failure Patterns
>  1. Yes. UFO already supports a three-level diagnosis. Tables 3–4 show when joint cues help or fail, Table 5 shows how cue quality differs, and Table 8 explains why performance does not improve.
>
> 2. The main pattern is consistent across these levels. The bottleneck is causal utility, not mere topical relevance. Some models generate cues that appear related to the task, but these cues are not sufficiently decision-critical to improve the final answer.
>
> 3. This is exactly our contribution. UFO diagnoses why joint reasoning fails, not only whether it fails. In the revision, we will add more annotated examples grouped by failure type (e.g., cue generation failure, cue–answer decoupling, and modality shortcut), with representative cases available at https://anonymous.4open.science/r/ICML_rebuttal-6B85/README.md.
>
> > ### Writing and State-Term Clarity.
>
> 1. We agree that the current state-first presentation creates unnecessary friction. In the revision, we will foreground task types as the reader-facing structure, use \textbf{state regimes} only as the unifying abstraction, and add a task-type/state-regime mapping table to improve readability.

---

> > ### Author Rebuttal · Reviewer_Zvyj · 2026-04-04
> >
> > Thanks for the authors' detailed responses, which successfully addressed my concerns. I have no further questions. Since my original rating is already positive, I'd like to maintain it.

---

> > > ### Author Response · Authors · 2026-04-07
> > >
> > > Dear Reviewer Zvyj:
> > >
> > > Thank you so much for the recognition of our responses. We are glad to receive your positive feedback!
> > > We are pleased to know that our rebuttal has fully addressed your concerns. We are deeply grateful for the professional guidance you provided throughout the review process.
> > >
> > > **Many thanks for your constructive comments, time, and patience.**
> > >
> > > Best regards and thanks,
> > >
> > > The Authors

---

### Decision · Program_Chairs · 2026-04-30

**Decision:**

Accept (regular)

**Comment:**

This paper proposes UFO, a benchmark for evaluating compositional multimodal reasoning in Unified Foundation Models (UFMs). It received mixed reviews: Weak Reject ×1, Weak Accept ×2, Accept ×1. The reviewers recognize that the idea of "evidential coupling" is a timely contribution, the benchmark is rigorously constructed, and the evaluation is comprehensive across 12 UFMs.

On the other hand, the reviewers raised concerns regarding the lack of clear justification for the necessity of joint reasoning, insufficient details about data construction, and the reliability of LLM-based evaluators. The authors provided a detailed rebuttal. Although one reviewer was not fully satisfied, most concerns have been addressed.

Considering the timely contribution of this work, I believe the strengths outweigh the weaknesses. Hence, I recommend accepting this paper.